# Comprehensive characterization of tumor microenvironment in colorectal cancer via molecular analysis

Xiangkun Wu[1,2†], Hong Yan[1,2,3†], Mingxing Qiu[1,2†], Xiaoping Qu[4,5†], Jing Wang[1,2], Shaowan Xu[1,2], Yiran Zheng[4,5], Minghui Ge[4,5], Linlin Yan[4,5], Li Liang[1,2,6*]

[1]Department of Pathology, Nanfang Hospital/School of Basic Medical Sciences, Southern Medical University, Guangzhou, China; [2]Department of Pathology and Guangdong Province Key Laboratory of Molecular Tumor Pathology, School of Basic Medical Sciences, Southern Medical University, Guangzhou, China; [3]Department of Pathology, The First Affiliated Hospital of USTC, Division of Life Sciences and Medicine, University of Science and Technology of China, Hefei, China; [4]Nanjing Simcere Medical Laboratory Science Co., Ltd, Nanjing, China; [5]State Key Laboratory of Translational Medicine and Innovative Drug Development, Jiangsu Simcere Diagnostics Co., Ltd, Nanjing, China; [6]Jinfeng Laboratory, Chongqing, China

*For correspondence: lli@smu.edu.cn

†These authors contributed equally to this work

**Abstract** Colorectal cancer (CRC) remains a challenging and deadly disease with high tumor microenvironment (TME) heterogeneity. Using an integrative multi-omics analysis and artificial intelligence-enabled spatial analysis of whole-slide images, we performed a comprehensive characterization of TME in colorectal cancer (CCCRC). CRC samples were classified into four CCCRC subtypes with distinct TME features, namely, C1 as the proliferative subtype with low immunogenicity; C2 as the immunosuppressed subtype with the terminally exhausted immune characteristics; C3 as the immune-excluded subtype with the distinct upregulation of stromal components and a lack of T cell infiltration in the tumor core; and C4 as the immunomodulatory subtype with the remarkable upregulation of anti-tumor immune components. The four CCCRC subtypes had distinct histopathologic and molecular characteristics, therapeutic efficacy, and prognosis. We found that the C1 subtype may be suitable for chemotherapy and cetuximab, the C2 subtype may benefit from a combination of chemotherapy and bevacizumab, the C3 subtype has increased sensitivity to the WNT pathway inhibitor WIKI4, and the C4 subtype is a potential candidate for immune checkpoint blockade treatment. Importantly, we established a simple gene classifier for accurate identification of each CCCRC subtype. Collectively our integrative analysis ultimately established a holistic framework to thoroughly dissect the TME of CRC, and the CCCRC classification system with high biological interpretability may contribute to biomarker discovery and future clinical trial design.

## Editor's evaluation

This study represents a valuable body of work in which the authors assemble a molecular description of colorectal cancer and a classification into subtypes. Overall, the evidence supporting the findings is solid, and consensus over a diverse range of data from publicly available sources is convincing. When added to existing knowledge this work may contribute to future biomarker discoveries for colorectal cancer.

## Introduction

Colorectal cancer (CRC) is the third most deadly malignancy worldwide (*Siegel et al., 2023*), and the incidence of early-onset CRC is steadily increasing (*Archambault et al., 2021*). CRC at early and localized stages is primarily a preventable and curable disease, but up to 50% of patients with locally advanced disease eventually develop mCRC (*Andrei et al., 2022*; *Ciardiello et al., 2022*). Therefore, the clinical systematic management of CRC patients is still an unmet medical challenge (*Ciardiello et al., 2022*).

With the development of high-throughput technologies and bioinformatics strategies, multi-omics data are used to identify and characterize the molecular subtypes of CRC, such as genomics (*Zhao et al., 2022*), transcriptomics (*Budinska et al., 2013*; *De Sousa E Melo et al., 2013*; *Marisa et al., 2013*; *Roepman et al., 2014*; *Sadanandam et al., 2013*; *Schlicker et al., 2012*), and proteomics (*Li et al., 2020*). The consensus molecular subtype (CMS) integrates six independent classification systems based on transcriptomics; however, it is still not explicitly used to guide clinical treatment (*Guinney et al., 2015*). The Cancer Genome Atlas Program (TCGA) and Clinical Proteomic Tumor Analysis Consortium (CPTAC) colorectal studies have dissected the molecular heterogeneity of CRC by integrating multi-omics data (*Cancer Genome Atlas Network, 2012*; *Vasaikar et al., 2019*). Nevertheless, multi-omics data are complex and highly dimensional, and extracting valuable information from these data to guide clinical treatment is still a tremendous challenge (*Leng et al., 2022*). By reviewing the biological characteristics of the tumor, useful information can be screened for identifying molecular subtypes.

The tumor cells can interact with cellular or non-cellular components, triggering dramatic molecular, cellular, and physical changes in the tumor microenvironment (TME) to build a self-sustainable tumor ecosystem (*Anderson and Simon, 2020*; *Chen and Song, 2022*). Simultaneously, TME profoundly affects tumor biology, responses to therapy, and clinical outcomes, which is a dynamic network mainly comprised of immune components and stromal components (*Hirata and Sahai, 2017*; *Jia et al., 2022*; *Zhang et al., 2022*). Furthermore, TME can adversely affect the metabolic activities of tumor, immune and stromal cells, and form diverse metabolic phenotypes (*Elia and Haigis, 2021*; *Kaymak et al., 2021*). Identifying the components of the TME and their functions, as well as the crosstalk between tumor cells and TME contributes to our understanding of the clinical heterogeneity of CRC, thereby bringing about new advances in precision medicine. Previous studies have used immune or stromal components of the TME, or a combination of both, to study the TME (*Bagaev et al., 2021*; *He et al., 2018*), but they are insufficient to completely reconstruct the heterogeneity of the TME.

In this study, we considered the tumor cells and its TME as a whole and performed a comprehensive characterization of TME in colorectal cancer (CCCRC), including the functional states of the tumor cells, immune and stromal signatures, and metabolic reprogramming features. We successfully identified the four CCCRC subtypes based on 61 TME-related signatures. Integrated analyses determined that the CCCRC subtypes had distinct histopathologic and molecular characteristics, therapeutic efficacy, and prognosis.

## Results

### Establishment of the TME panel

The molecular and clinical features of a tumor are characterized by the functional states of tumor cells, as well as other TME-related signatures, including immune and stromal components, and metabolic reprogramming signatures. In brief, 15 signatures (including angiogenesis, apoptosis, cell cycle, differentiation, DNA damage, DNA repair, endothelial-to-mesenchymal transition (EMT), hypoxia, inflammation, invasion, metastasis, proliferation, quiescence, stemness, and cancer stem cells) were used to describe the functional states of tumor cells. As for the immune signatures, we focused on eight categories of immune cells (T cells, natural killer cells, dendritic cells, macrophages, myeloid-derived suppressor cells, B cells, mast cells, and neutrophils) and their subpopulations, as well as the other immune-related signatures. In addition to the signatures of endothelial cells, mesenchymal cells, and mesenchymal stem cells, we included signatures of extracellular matrix, matrix remodeling and interactions of cells with the extracellular matrix to characterize the stromal compartments. A total of seven major metabolic pathways (amino acid, nucleotide, vitamin cofactor, carbohydrate, TCA cycle, energy, and lipid metabolism) were used to reveal the metabolic reprogramming of the TME. According to

the above biological framework, a total of 61 TME-related signatures were collected to form the TME panel (*Supplementary file 1a*, *Supplementary file 1b*), which ultimately established a holistic approach to thoroughly dissect the TME of CRC.

Principal component analysis (PCA) indicated that after using the 'ComBat' function to remove batch effects, there was no significant batch effect in the merged cohorts of eight microarray datasets (*Figure 1—figure supplement 1A, B*) and two RNA sequencing datasets (*Figure 1—figure supplement 1C, D*, *Supplementary file 1c*). We used Gene set variation analysis (GSVA) to calculate the TME-related signature scores for each sample in each cohort. Principal coordinate analysis (PCOA) revealed that the CRC samples could be distinguished from normal samples by the TME-related signatures in the GSE39582 and TCGA cohorts (*Figure 1—figure supplement 1E*). We further focused on the signatures of the functional states of tumor cells, which could classify CRC and normal samples (*Figure 1—figure supplement 1F*). The p-values for intercomparisons of the 'Euclidean' distances between normal and CRC samples were all <0.05 using permutational multivariate analysis of variance (PERMANOVA) test (*Zhu et al., 2021*). Most immune signatures had higher GSVA scores in the normal samples compared with the CRC samples, while stromal signatures and the signatures of the functional states of tumor cells had higher GSVA scores in CRC tissues (*Figure 1—figure supplement 1G, H*). As expected, amino acid, carbohydrate, and nucleotide metabolic processes were more prominent in CRC samples, which was consistent with the hallmark of infinite proliferation of tumor cells (*Figure 1—figure supplement 1G, H*).

Spearman's correlation analysis of the TME-related signatures revealed three major patterns bound by positive correlations in the CRC-AFFY cohort (*Figure 1—figure supplement 1I*). One pattern defining the proliferation of tumor cells consisted of cell cycle and metabolic reprogramming signatures. The second was mainly comprised of immune components, such as T cells, natural killer cells (NK cells), myeloid-derived suppressor cells (MDSCs), and M2 macrophages. The third pattern was associated with stromal components such as angiogenesis and extracellular matrix, as well mesenchymal cells and cancer stem cells. In addition, our findings indicated a strong positive correlation between lymphocytic and stromal signatures and MCP-counter algorithm-derived signatures, thereby emphasizing the robustness of the employed methods (*Figure 1—figure supplement 1J*). Meanwhile, we found that 15 signatures associated with the functional states of tumor cells were positively correlated with the activity of 10 classical oncogenic pathways (*Figure 1—figure supplement 1K*). Finally, we used the Kaplan–Meier method and univariate Cox proportional hazard regression analysis to evaluate the prognosis of the TME-related signatures. The stromal and tumor components significantly were correlated with decreased survival, particularly in the case of mesenchymal cells, endothelial cells, metastasis, differentiation, and EMT signatures (*Figure 1—figure supplement 1L*, *Supplementary file 1d*). Lymphoid-associated cells generally tended to be associated with a better prognosis, while myeloid-associated cells generally tended to be associated with a poor prognosis. Among the metabolism-related signatures, energy metabolism and carbohydrate metabolism were significantly related to poor prognosis, while nucleotide metabolism, amino acid metabolism, and TCA metabolism were strongly predictive of a favorable prognosis. Collectively, our findings demonstrated that the TME heterogeneity, including unique differences in immune, stromal, and metabolic reprogramming, played a crucial role in tumor development, and that the TME panel could be used to comprehensively characterize CRC.

## Identification and validation of CCCRC classification

With the increasing application of immunotherapy and tumor vaccines, there is growing evidence highlighting the importance of the TME in tumorigenesis and development (*Bejarano et al., 2021*; *Saxena et al., 2021*). To reveal the TME heterogeneity of CRC using the curated TME panel, consensus clustering analysis was performed based on the 61 TME-related signature scores in the CRC-AFFY cohort, and the optimal cluster number was determined to be four using the consensus matrices heatmap, the empirical cumulative distribution function (CDF) plot, and delta area plot (*Wilkerson and Hayes, 2010*) (Materials and methods, *Figure 1—figure supplement 2A–C*). Subsequently, the CRC samples in the CRC-AFFY cohort were classified into the four CCCRCs with distinct TME components (*Figure 1A, B*, *Figure 1—figure supplement 2D*). To evaluate the reproducibility of the CCCRC subtypes, we utilized the PAM (Prediction Analysis of Microarrays) algorithm (*Tibshirani et al., 2002*) to extract 61 TME-related signatures that best represent each subtype, using a threshold of 0.566

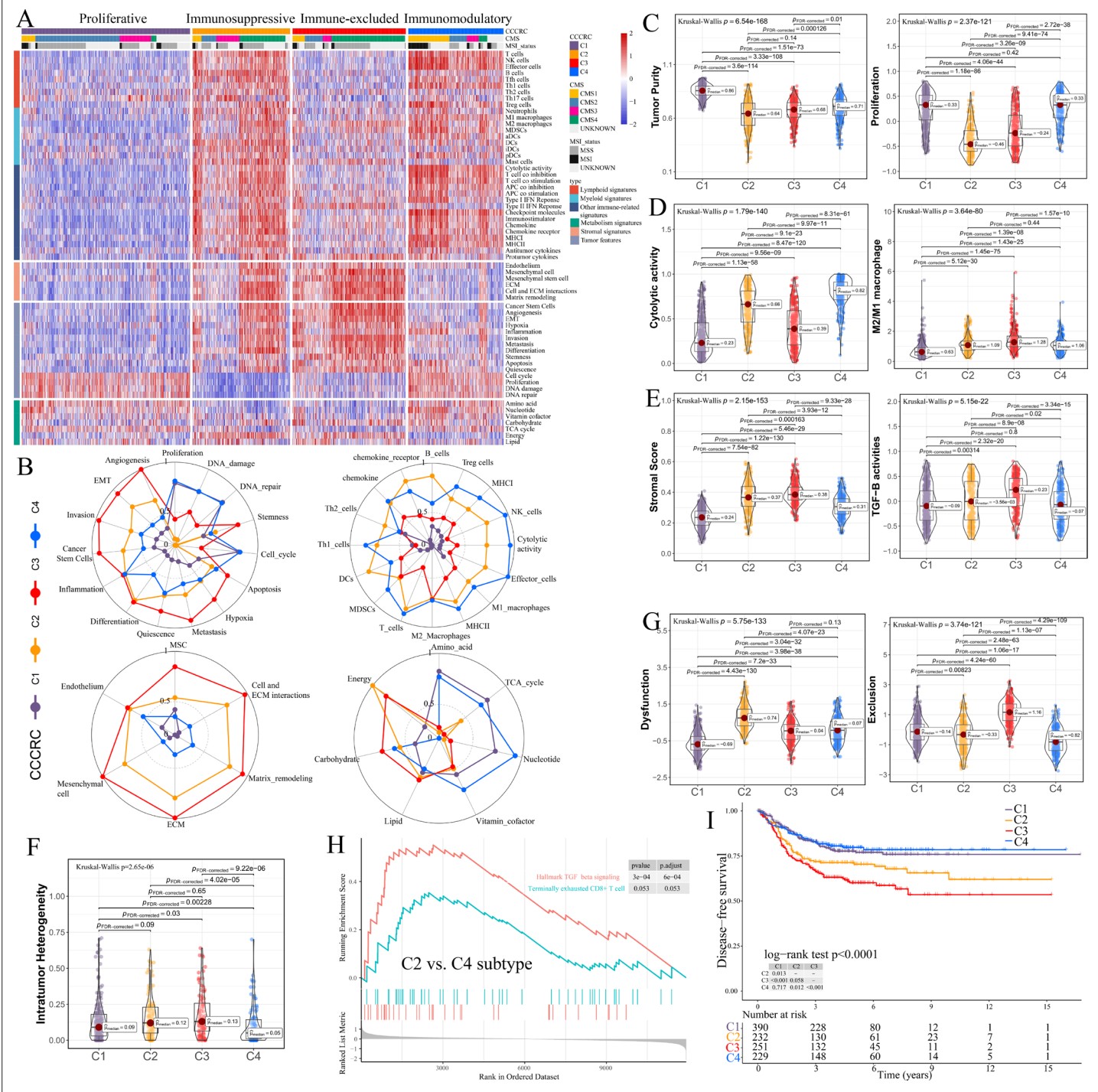

**Figure 1.** Comprehensive characterization of colorectal cancer (CCCRC). (**A**) Heatmap of 1471 colorectal cancer (CRC) patients in the CRC-AFFY cohort classified into four distinct tumor microenvironment (TME) subtypes based on the 61 TME-related signatures. CMS: consensus molecular subtypes; MSI: microsatellite instability; MSS: microsatellite stability. (**B**) Radars display the characteristic TME-related signatures, including tumor, immune, stroma, and metabolism signatures, of each CCCRC subtype in the CRC-AFFY cohort. MSC: mesenchymal stem cells; ECM: extracellular rmatrix. Box plots show differences in tumor (**C**), immune (**D**), and stroma (**E**) signatures in the CRC-AFFY cohort. Tumor purity and stroma scores were obtained from the ESTIMATE algorithm. Proliferative activity (proliferation), cytolytic score, M1 and M2 macrophage proportions, and TGFB activity were calculated by Gene set variation analysis (GSVA). (**F**) Differences in intratumor heterogeneity between four CCCRC subtypes in CRC-RNAseq cohort. (**G**) Differences in T cell dysfunction and T cell exclusion scores between four CCCRC subtypes were analyzed based on the gene expression profiles in CRC-AFFY cohort. (**H**) Gene set enrichment analysis (GSEA) of the terminally exhausted CD8+ T cell signature and the TGF-beta signaling signature between C2 and C4

*Figure 1 continued on next page*

*Figure 1 continued*

subtypes in the CRC-AFFY cohort. (**I**) Kaplan–Meier method with log-rank test of disease-free survival among the four CCCRC subtypes in the CRC-AFFY cohort.

The online version of this article includes the following figure supplement(s) for figure 1:

**Figure supplement 1.** Establishment of tumor microenvironment (TME) panel.

**Figure supplement 2.** Comprehensive characterization of colorectal cancer (CCCRC).

**Figure supplement 3.** Differences in the tumor microenvironment (TME) components obtained from other algorithm among the comprehensive characterization of colorectal cancer (CCCRC) subtypes.

**Figure supplement 4.** Limiting the consensus clustering analysis to only immune-related or immune- and stroma-related signatures in the CRC-AFFY cohort.

**Figure supplement 5.** Overlap of the comprehensive characterization of colorectal cancer (CCCRC) subtypes with published colorectal cancer (CRC) molecular subtypes in the CRC-AFFY and CRC-RNAseq cohorts.

**Figure supplement 6.** Associations between comprehensive characterization of colorectal cancer (CCCRC) subtypes and consensus molecular subtypes (CMS).

**Figure supplement 7.** Survival analyses of the comprehensive characterization of colorectal cancer (CCCRC) subtypes and consensus molecular subtypes (CMS).

(*Figure 1—figure supplement 2E*). These signatures were then used to construct a PAMR classifier with superior predictive capability, exhibiting an overall error rate of 15%. The PAMR classifier based on the PAM algorithm is publicly available at https://github.com/XiangkunWu/PAMR_classifier (copy archived at *Wu, 2023a*). We used the established PAMR classifier to predict the CCCRC subtypes on the CRC-RNAseq cohort and the same four CCCRC subtypes were revealed, with similar patterns of differences in the TME components (*Figure 1—figure supplement 2F, G*). PCOA showed that the four CCCRC subtypes were distinctly separated and the p-values for intercomparisons of the 'Euclidean' distances between them were all <0.05 using PERMANOVA test (*Figure 1—figure supplement 2H*) in the CRC-RNAseq and CRC-AFFY cohorts. PCOA also demonstrated highly similar TME compartments in the same subtype between the CRC-RNAseq and CRC-AFFY cohorts. Differences in the TME components between the CCCRC subtypes were also observed in the analysis of previously reported immune and stromal signatures obtained by the microenvironment cell populations (MCP)-counter, CIBERSORT, and ESTIMATE algorithm, and 10 classical oncogenic pathway activities and 86 metabolic pathway enrichment scores calculated by GSVA (*Figure 1—figure supplement 3A–E*, *Supplementary file 1e*). Notably, limiting the consensus clustering analysis to only immune-related or immune- and stroma-related signatures, as done in previous studies (*Bagaev et al., 2021*; *He et al., 2018*), did not allow reliable identification of all four CCCRC subtypes (*Figure 1—figure supplement 4*). These sensitivity analyses underscored the necessity of our well-designed TME panel to achieve this identification.

C1 (35% of all tumors), hereafter designated as the proliferative subtype, was characterized by the relative upregulation of tumor proliferative activity, tumor purity, and minimal or complete lack of lymphocyte and stromal infiltration, which was highly similar to the cold tumor phenotype (*Figure 1B–E*). The MYC, cell cycle, TP53, and PI3K pathways associated with tumor proliferation had the highest GSVA scores in the C1 subtype (*Figure 1—figure supplement 3E*). C2 (21% of all tumors), hereafter designated as the immunosuppressed subtype, was characterized by the relative upregulation of immune and stromal components, such as T cells, M2 macrophages, and cancer-associated fibroblasts (CAFs) (*Figure 1B–E*, *Figure 1—figure supplement 3A–D*). However, the extent of infiltration of effector cells, as well as the cytolytic score, was much lower than that of the C4 subtype. C3 (24% of all tumors), hereafter designated as the immune-excluded subtype, was characterized by the distinct upregulation of stromal components, such as CAFs, and cancer stem cells, as well as angiogenesis and hypoxia signatures (*Figure 1B–E*, *Figure 1—figure supplement 3A–D*). During tumor progression, TGF-beta secreted by CAFs is leveraged by tumor cells to suppress and exclude the anti-tumor immune components (*Liu et al., 2019*). We observed that the TGF-beta pathway, as well as WNT, NOTCH, and RAS pathways, and the ratio of M2/M1 macrophages, were distinctly upregulated in C2 and C3 subtypes (*Figure 1D, E*, *Figure 1—figure supplement 3E*). The scores of 5/10 oncogenic pathways were the highest in the C3 subtype (*Figure 1—figure supplement 3E*), suggesting that the activation of oncogenic pathways could lead to the formation of immune-excluded phenotypes which

was consistent with the previous theory (*Galon and Bruni, 2019*). The level of intratumor heterogeneity (ITH) was significantly linked to poor prognosis and drug resistance (*Caswell and Swanton, 2017*). As expected, the ITH of the C2 and C3 subtypes was higher than that of the other subtypes (*Figure 1F*). C4 (20% of all tumors), hereafter designated as the immunomodulatory subtype, was characterized by the remarkable upregulation of anti-tumor immune components, such as effector T cells, NK cells, and Th1 cells. The C4 subtype also had the highest cytolytic score compared with the other subtypes and lacked stromal components and the other immunosuppressed components, which indicated an immunomodulatory microenvironment (*Figure 1B–E*).

To further explore the immune escape mechanism of each CCCRC subtype, the differences in T cell dysfunction and T cell exclusion scores between the four CCCRC subtypes were analyzed based on the gene expression profiles (GEP), which reflected the T cell function of the global tumor (*Jiang et al., 2018*). Strikingly, the C2 subtype had highest T cell dysfunction score, indicating that T cell exhaustion in the C2 subtype was at the late stage (*Figure 1G*, *Figure 1—figure supplement 3F*). Using gene set enrichment analysis (GSEA) with all genes ranked according to the fold change (FC) between C2 and C4 subtypes, we found that terminally exhausted CD8+ T cell and TGF-beta signaling signatures were upregulated in the C2 subtype in the CRC-AFFY (*Figure 1H*) and CRC-RNAseq (*Figure 1—figure supplement 3G*) cohorts, which might reveal that CD8+ T cell infiltration within the tumor bed was suppressed by the stroma and was in a late state of exhaustion. The C3 subtype had the highest T cell exclusion score (*Figure 1G*, *Figure 1—figure supplement 3F*), demonstrating that the low T cell infiltration into the tumor bed was due to the increased abundance of CAFs and M2 macrophages, thereby leading to the exclusion of T cells from the tumor bed. Metabolic reprogramming also differed significantly among the four CCCRC subtypes (*Figure 1B*, *Figure 1—figure supplement 3H*). We analyzed the 86 metabolic pathways obtained from the Kyoto Encyclopedia of Genes and Genomes (KEGG) database (*Supplementary file 1e*) and observed that the number of upregulated metabolic pathways of the C3 subtype was the lowest. We also found that glycan metabolism was distinctly upregulated in C2 and C3 subtypes, which indicated that glycan metabolism was significantly associated with the stroma.

## Associations between CCCRC subtypes and other molecular subtypes and clinical characteristics

Previous studies have identified several molecular subtypes of CRC based on GEP (*Budinska et al., 2013*; *De Sousa E Melo et al., 2013*; *Guinney et al., 2015*; *Roepman et al., 2014*; *Sadanandam et al., 2013*). We investigated their associations with the CCCRC subtypes in the CRC-AFFY and CRC-RNAseq cohorts (*Figure 1—figure supplement 5*). The C1 subtype was primarily comprised of the CMS2 and lower crypt-like subtypes, and it contained the highest frequencies of the CCS1, B-type, and TA subtypes. The C2 subtype mainly consisted of the CMS4, stem-like, surface crypt-like, CCS3, and C-type subtypes, and included the highest frequency of the enterocyte subtype. The C3 subtype contained the highest frequencies of CMS4, stem-like, CCS3, and C-type subtypes and was mainly comprised of the mesenchymal and TA subtypes. The C4 subtype included the highest frequencies of microsatellite instability (MSI) and the CMS1, CIMP-H-like, A-type, and inflammatory subtypes, and was mainly comprised of the CCS2 subtype.

We also focused on the differences in the TME components between the CCCRC and the CMS subtypes. Compared with the CMS1 subtype, the C4 subtype showed upregulated anti-tumor immune components and lacked immunosuppressive components (*Figure 1—figure supplement 6A*). CRC patients with MSI were sensitive to immune checkpoint blockade (ICB) therapy (*Jin and Sinicrope, 2022*), and C4 and CMS1 subtypes containing approximately 47 and 75% of MSI cases, respectively. The C4 subtype with MSI showed upregulated scores of effector cells and cytolytic activity and downregulated scores of extracellular matrix and matrix remodeling compared with the CMS1 subtype with MSI (*Figure 1—figure supplement 6B*). Moreover, we observed that the C4 subtype with MSI and the C4 subtype with MSS had higher scores of anti-tumor immune signatures and lower scores of stromal components than the other subtype with MSI (*Figure 1—figure supplement 6C*). We also observed that CMS2 subtype contained more C4 subtypes in addition to mainly C1 subtypes. Therefore, we analyzed the differences in the TME components between C1 and CMS2 subtypes and found that CMS2 subtypes indeed had higher immune-related components than C1 subtypes, such as MCH-I, MCH-II, and inflammatory signature, and also contained more stromal components, such

as extracellular matrix, than C1 subtypes (*Figure 1—figure supplement 6D*). This suggested that the C1 subtype was less immunogenic than CMS2 and more closely resembles cold tumor characteristics. Specifically, we found that CMS4 contained mainly C2 and C3 subtypes. Our findings indicated that the C2 subtypes within CMS4 exhibited a higher abundance of immune components, such as T cells and NK cells, compared to the C3 subtype within CMS4. However, the differences in stromal components between these subtypes were not statistically significant (*Figure 1—figure supplement 6E*).

We further analyzed the association of CCCRC subtypes with clinicopathological characteristics (*Supplementary file 1f*, *Supplementary file 1g*). We found that the C4 subtype was mostly diagnosed in right-sided CRC lesions and in females, which was consistent with the CMS1 subtype. The C1 and C3 subtypes were mainly observed in left-sided CRC lesions and in males, consistent with the CMS2 and CMS4 subtypes. The C3 subtype was strongly associated with more advanced tumor stages, which was the similarity to the CMS4 subtype, while the C4 subtype was associated with higher histopathologic grade, which was the similarity to the CMS1 subtype. Furthermore, our analysis using the Kaplan–Meier method demonstrated that patients with the C4 subtype had significantly higher disease-free survival (DFS) and overall survival (OS) compared to those with the C2 and C3 subtypes in the CRC-AFFY (*Figure 1I*, *Figure 1—figure supplement 7A*) and CRC-RNAseq cohorts (*Figure 1—figure supplement 7B, C*). Multivariate Cox proportional hazard regression analysis showed that the C4 subtype was an independent predictor of the best OS and DFS, whereas the C3 subtype was an independent predictor of the worst OS and DFS after adjustment for age, gender, tumor site, TNM stage, grade, adjuvant chemotherapy or not, MSI status, BRAF and KRAS mutations, and the CMS classification system in the combined cohort (the CRC-AFFY and CRC-RNAseq cohorts) (*Supplementary file 1h*). Considering that the C1, C2/C3, and C4 subtypes partially overlap with the CMS2, CMS4, and CMS1 subtypes, respectively, we also analyzed the prognostic differences between them in the combined cohort. We found that the DFS/OS of patients with the C1 subtype was worse than those with the CMS2 subtype (*Figure 1—figure supplement 7D, E*), the DFS/OS of patients with the C2 subtype was better than those with the CMS4 subtype (*Figure 1—figure supplement 7F, G*), the DFS/OS of patients with the C3 subtype was not significantly different from those with the CMS4 subtype (*Figure 1—figure supplement 7F, G*), and the DFS/OS of patients with the C4 subtype was significantly better than those with the CMS1 subtype (*Figure 1—figure supplement 7H, I*). Notably, the C2 subtype within the CMS4 subtype also had a better prognosis than the C3 subtype within the CMS4 subtype (*Figure 1—figure supplement 7J, K*). The above analysis demonstrated that the CCCRC classification system was closely associated with clinicopathological characteristics, were able to refine the CMS classification system and MSI status, as well as contributed to the understanding of the mechanisms underlying the different clinical phenotypes resulting from TME heterogeneity.

## Differences in histopathologic characteristics between CCCRC subtypes

To further explore the biological differences between CCCRC subtypes, we investigated the histopathologic phenotypes by evaluating the whole-slide images (WSIs) of the TCGA-CRC dataset. We compared our CCCRC system with the three-category immune classification system of solid tumors, termed 'desert', 'excluded', and 'inflamed' phenotypes (*Chen and Mellman, 2017*; *Rosenberg et al., 2016*). Two pathologists evaluated the histopathologic characteristics for each subtype under the microscope. The 254 CRC samples in the TCGA-CRC dataset were categorized as these three phenotypes based on the abundance of lymphocytes and their spatial location with malignant epithelial cells (*Supplementary file 1i*). According to the three-category immune classification system, the C4 subtype was enriched with an inflamed phenotype characterized by abundant lymphocytes in direct contact with malignant cells (*Figure 2A*). The C2 subtype was mostly categorized as an excluded phenotype. The C1 and C3 subtypes were mainly classified into the desert phenotype, whereas the C3 subtype was more frequently classified as an excluded phenotype than the C1 subtype. Notably, the lymphocytes of C2 subtype were more frequently intermixed with intratumor stromal components, whereas the lymphocytes of C3 subtype were more frequently excluded from the tumor bed and intermixed with adjacent-tumor stromal components, both of which were classified as excluded phenotype according to the three-category immune classification system.

The above differences in the histopathologic characteristics among the CCCRC subtypes were based on the semi-quantitative analysis results of two pathologists, which are subjective to a certain extent. Therefore, we used hematoxylin and eosin (HE)-stained image-based deep learning to establish the

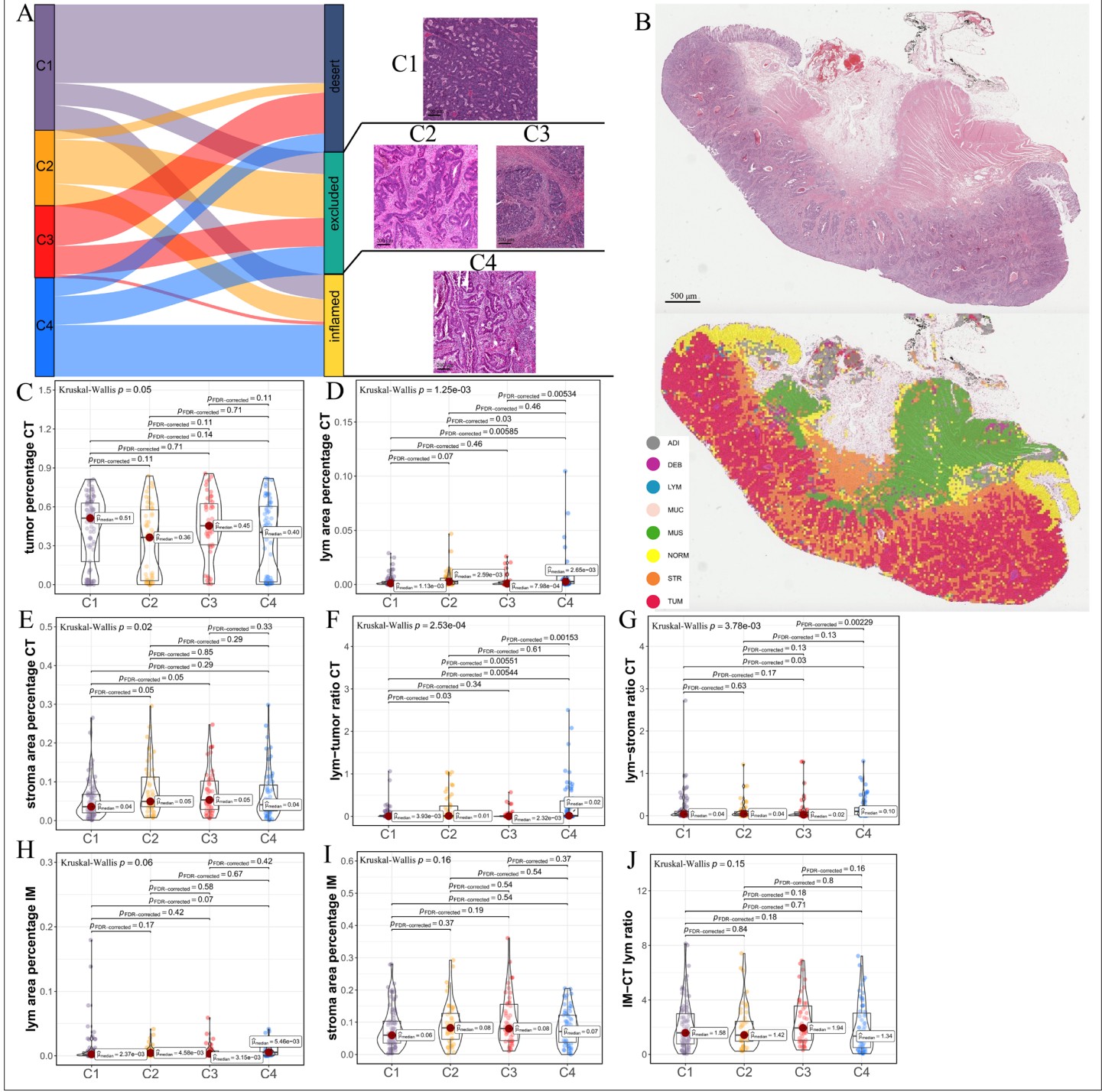

**Figure 2.** Differences in histological characteristics between comprehensive characterization of colorectal cancer (CCCRC) subtypes. (**A**) Sankey plot shows overlap of CCCRC subtypes with the three-category immune classification system ('desert', 'excluded', and 'inflamed' phenotypes), and their representative hematoxylin and eosin (HE)-stained whole-slide images (WSIs). C1: TCGA-AA-3955; C2: TCGA-A6-6654; C3: TCGA-CK-4948; and C4: TCGA-AD-6963. Scale bars 200 μm.(**B**) Representative WSI (top) and the colorectal cancer (CRC)-multiclass model-inference segmentation of eight tissue types: tumor, stroma, lymphocyte, normal colon mucosa, debris, adipose, mucin, and muscle (bottom). Scale bar 500 μm. Box plots show differences in the abundance of tumors (**C**), lymphocyte infiltration (lym) (**D**), and stroma (**E**) in the core tumor (CT) region. Box plots show differences in the lymphocyte infiltration to tumor content ratio (**F**) and lymphocyte infiltration to stromal content ratio (**G**) in the CT region. Box plots show differences in the abundance of lymphocytes infiltration (**H**) and stroma (**I**) in the invasive margin (IM) region. (**J**) Box plots show differences in the ratio of lymphocyte infiltration in the IM region to the CT region.

*Figure 2 continued on next page*

*Figure 2 continued*

The online version of this article includes the following figure supplement(s) for figure 2:

**Figure supplement 1.** The performance of the colorectal cancer (CRC)-multiclass model: a muscle/non-muscle classifier a seven-tissue classifier.

CRC-multiclass model and to evaluate the abundance and spatial distribution of the tumor, lympho-cytes, and stroma. The CRC-multiclass model consisted of a muscle/non-muscle classifier and a seven-tissue classifier that could classify eight CRC tissue types: adipose (ADI), debris (DEB), lymphocytes (LYM), mucus (MUC), smooth muscle (MUS), normal colon mucosa (NORM), cancer-associated stroma (STR), and colorectal adenocarcinoma epithelium (TUM) (Materials and methods). The muscle/non-muscle model demonstrated well performance on the internal test set of 14,681 patches, achieving an area under the curve (AUC) of approximately 0.99 and an accuracy of 0.99 (*Figure 2—figure supplement 1A, B*). Meanwhile, the seven-tissue classifier also performed well, with AUCs for the different tissue types above 0.99 and an accuracy of 0.95 on the internal test set of 5741 patches (*Figure 2—figure supplement 1C, D*). When evaluated on an external test set of 4288 patches, the muscle/non-muscle model achieved an AUC of 0.95 and an accuracy of 0.91 (*Figure 2—figure supplement 1E, F*). Of the 3633 patches identified as non-muscle by the muscle/non-muscle model, the seven-tissue classifier achieved AUCs ranging from 0.97 to 1 for different tissue types and an accuracy of 0.91 (*Figure 2—figure supplement 1G, H*). The tissue heatmap showed our model prediction results for a TCGA-CRC WSI (*Figure 2B*). In the core tumor (CT) region, the C1 subtype had a highly increased abundance of the tumor; the C4 subtype had increased lymphocyte infiltration and decreased stromal content; the C2 subtype had elevated lymphocyte and stromal infiltration; and the C3 subtype had the highest abundance of stroma, but less lymphocyte infiltration was detected (*Figure 2C–E*). We also observed that C4 subtype had the highest lymphocyte infiltration to tumor content ratio and lympho-cyte infiltration to stromal content ratio, followed by C2 subtype and C3 subtype had the lowest (*Figure 2F, G*). In the invasive margin (IM) region, different degrees of lymphocyte infiltration and stromal components were observed for each subtype (*Figure 2H,I*). Importantly, the ratio of lympho-cyte infiltration in the IM region of the C3 subtype to the CT region was the highest, which confirmed that the stromal components excluded lymphocytes from the CT region in the C3 subtype (*Figure 2J*). AI-enabled spatial analysis of WSIs confirmed the semi-quantitative results of the pathologists, with the C1 subtype belonging to the cold phenotype, C2 subtype belonging to the immunosuppressive phenotype, C3 subtype belonging to the excluded phenotype, and C4 subtype belonging to the hot phenotype. Collectively, our CCCRC system further refined the three-category immune classification system of solid tumors (*Chen and Mellman, 2017*; *Rosenberg et al., 2016*) and conformed to the four-category immune classification system, termed 'hot', 'cold', 'immune-excluded', and 'immuno-suppressive' phenotypes (*Galon and Bruni, 2019*; *Kirchhammer et al., 2022*).

## Biological characterization of CCCRC subtypes

We further elucidated the differences in biological characteristics among the CCCRC subtypes using multi-omics data, including genomics, epigenetics, transcriptomics, and proteomics data. As for the genomic alterations, the C4 subtype had the highest TMB and neoantigen values and the lowest prevalence of chromosomal instability (CIN), including somatic copy number alteration (SCNA) counts and fraction of the genome altered (FGA) scores, compared with the other subtypes (*Figure 3A, B*). Conversely, C1 and C3 subtypes displayed the highest CIN levels, as described by SCNA counts and FGA scores, and the lowest TMB and neoantigen values (*Figure 3A, B*). The C2 subtype displayed median CIN levels, TMB and neoantigen values. Among the frequently mutated genes (>5%), the mutation frequencies of *APC* (85.8%), *TP53* (64.9%), and *KRAS* (46.7%) were the highest in the C1 subtype compared to the other subtypes (all p-value <0.05), followed by the C3, C2, and C4 subtypes, which are closely associated with the occurrence of CRC (*Figure 3A*, *Supplementary file 2a*). The C4 subtype was significantly enriched in mutations of *DNAH2* (26.0%), *MYH8* (26.8%), and *BRAF* (26.0%) genes (all p-value <0.05), whereas the mutation frequency of C1, C2, and C3 subtypes was low. In terms of the differences in SCNA, the C1 subtype with the highest CIN level harbored significantly more amplified chromosomal regions (20q12, 20q13.12, 20q11.21, and 20q13.32) and deleted chro-mosomal regions (18q21.2, 18q22.1, and 18q12.3) (all p-value <0.05) (*Figure 3A, B*, *Supplementary file 2b*). The C3 subtype was significantly enriched in the amplified chromosomal regions of 13q33.3, 13q22.1, and 13q12.2 and the deleted chromosomal regions of 8p21.2 and 8p23.2 (all, p-value <0.05).

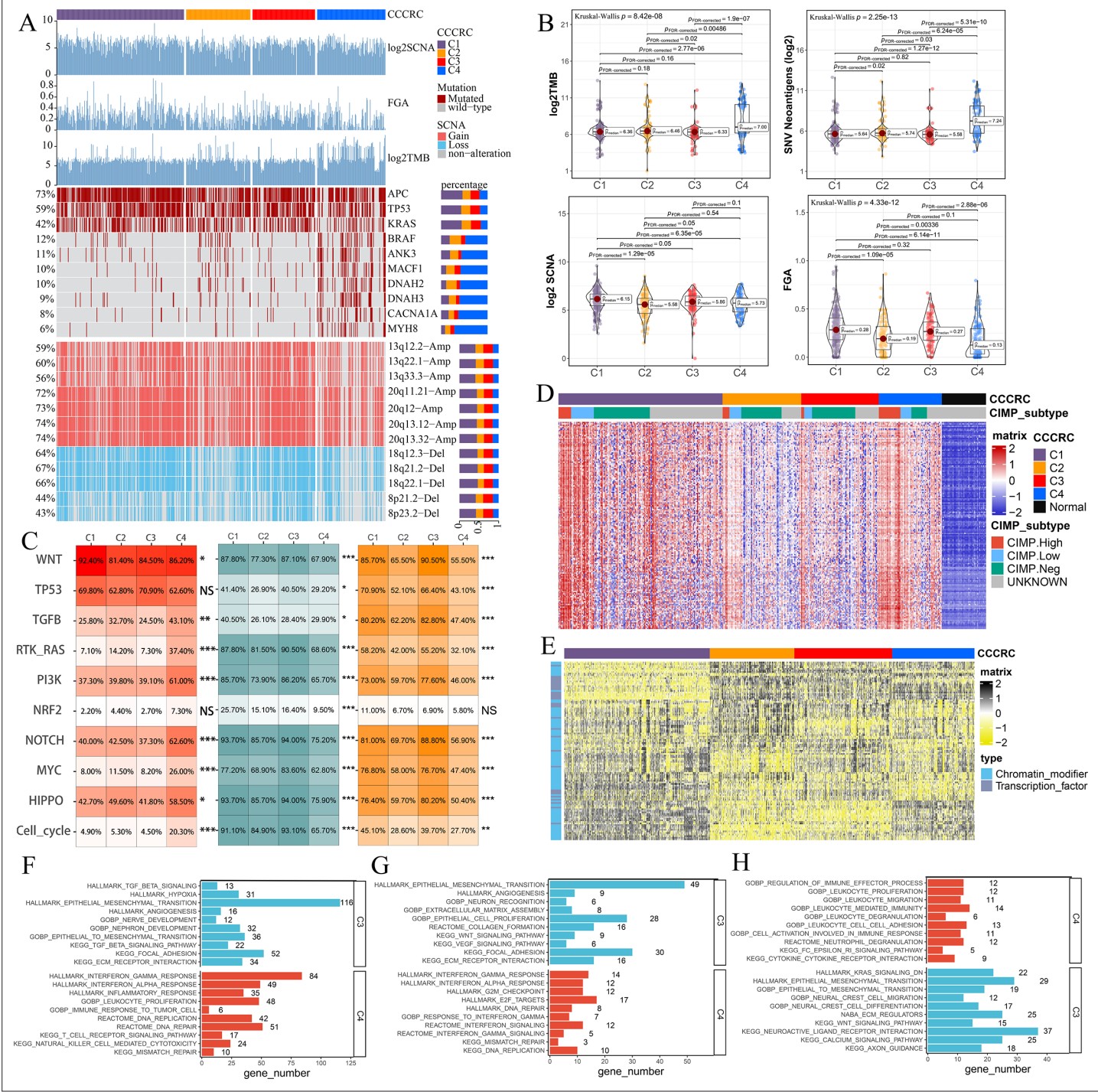

**Figure 3.** Biological characterization of comprehensive characterization of colorectal cancer (CCCRC) subtypes based on multi-omics data. (**A**) Distribution of driver gene mutations and somatic copy number alterations (SCNAs) among the CCCRC subtypes in the TCGA-CRC dataset. (**B**) Box plots show differences in tumor mutation burden (TMB), neoantigens, SCNA counts, and fraction of the genome altered (FGA) scores among the four CCCRC subtypes in the TCGA-CRC dataset. (**C**) Genomic alterations in 10 oncogenic pathways were compared among the four CCCRC subtypes in the TCGA-CRC dataset. The color of the box represents the different types of genomic alterations (red, mutation; blue, amplification; yellow, deletion), and the color saturation represents the frequency. The color of the p-value represents which oncogenic pathway had the highest frequency of the genomic alterations. Heatmap shows differentially methylated genes derived from each CCCRC subtype versus normal tissues (**D**) and regulon activity profiles for transcription factors and chromatin modifiers (**E**). (**F**) Significantly enriched gene sets among genes upregulated in the C4 subtype (red bars) and the C3 subtype (blue bars). (**G**) Significantly enriched gene sets among proteins upregulated in the C4 subtype (red bars) and the C3 subtype (blue bars). (**H**) Significantly enriched gene sets of methylated genes with downregulated DNA methylation in the C4 subtype compared to the C3 subtype (red bars) or

*Figure 3 continued on next page*

*Figure 3 continued*

with upregulated DNA methylation in the C4 subtype compared to the C3 subtype (blue bars). *p-value <0.05; **p-value <0.01; ***p-value <0.001; NS, p-value >0.05.

The online version of this article includes the following figure supplement(s) for figure 3:

**Figure supplement 1.** Biological characterization of the comprehensive characterization of colorectal cancer (CCCRC) subtypes based on multi-omics data.

No SCNA was significantly enriched in C2 and C4 subtypes. The single alteration events could not adequately delineate the CCCRC subtypes, we further computed the fraction of the altered samples per oncogenic pathway in each CCCRC subtype. The C4 subtype had the highest frequency of mutations in the cell cycle, HIPPO, MYC, NOTCH, PI3K, TGFB, and RTK–RAS pathways (all p-value <0.05) (*Figure 3C*, *Supplementary file 2c*). Notably, the C1 subtype had the highest frequency of mutations in the WNT pathway (p-value = 0.019). The frequency of mutations in the TP53 pathway was not significantly different between CCCRC subtypes. The 10 oncogenic pathways had higher frequencies of amplification (all p-value <0.05), and 9 oncogenic pathways (except the NRF2 pathway) had higher frequencies of deletion (all p-value <0.05) in C1 and C3 subtypes compared with C2 and C4 subtypes. Although none of genomic alterations was limited to or specific to a particular subtype, the apparent enrichment of certain alteration events within the CCCRC subtypes might highlight the TME heterogeneity and the genotype-CCCRC correlations of CRC.

Subsequently, we found that the different CCCRC subtypes displayed highly diverse epigenetic, transcriptional, and proteomic profiles. As expected, the analysis of differentially methylated genes (DMGs) between CRC and normal tissues demonstrated that the C4 subtype had the most DMGs (*n* = 145) cared to the C1 subtype (*n* = 109), C2 subtype (*n* = 12), and C3 subtype (*n* = 23), and the C4 subtype exhibited extensive hypermethylation with the highest frequency of the CpG island methylator phenotype (CIMP) compared with the other subtypes (*Figure 3D*). We further analyzed the regulon activity of critical chromatin modifiers and transcription factors in CRC, which could better evaluate their combinatorial biological effects. The regulon activity of the chromatin modifiers of the C1 subtype was generally higher than that of the other subtypes (*Figure 3E*). The differences in the regulon activity of the chromatin modifiers might indicate that epigenetically driven transcriptional networks contributed to the remodeling of the TME, especially in the C1 subtype. Meanwhile, we observed that each subtype had different transcription factor activities (*Figure 3E*). As shown in *Figure 3—figure supplement 1A*, C1-specific upregulated genes (false discovery rate [FDR]<0.001, top 1000 by log2FC) were enriched for the pathways associated with tumor proliferation and metabolism. C2-specific upregulated genes were enriched for the pathways associated with immune function, stroma, and neurons. C3-specific upregulated genes were enriched for the pathways associated with stroma and neurons. Both C2- and C3-specific upregulated genes were enriched in neuron-associated pathways, suggesting that neuronal development might be involved in the formation of ECM. C4-specific upregulated genes were enriched for the pathways associated with anti-tumor immune function. The lists of differentially expressed genes (DEGs) between CCCRC subtypes were compiled in *Supplementary file 2d*. In addition, we also analyzed the correlations between all genes and the exclusion and dysfunction scores obtained from the Tumor Immune Dysfunction and Exclusion (TIDE) website, as well as the MSI expression signature, which provided a useful reference for subsequent research in the discovery of new therapeutic targets (*Supplementary file 2e*). The CCCRC-specific upregulated methylation genes (FDR <0.001, top 1000 by difference) and the CCCRC-specific upregulated proteins (p-value <0.05) were also enriched for analogous biological functional categories (*Figure 3—figure supplement 1B, C*). Gene expression differences among the CCCRC subtypes were validated in the CRC-RNAseq cohort (*Figure 3—figure supplement 1D–G*). DMGs (FDR <0.001, top 1000 by log2FC), DEGs (FDR <0.001, top 1000 by log2FC), and differentially expressed proteins (DEPs) (p-value <0.05) between each subtype were enriched for similar biological functional categories. Indeed, DEGs and DEPs upregulated in the C4 subtype compared with the C3 subtype were significantly enriched for immune-related pathways, whereas DEGs and DEPs upregulated in the C3 subtype compared with the C4 subtype were highly enriched for TGF-beta signaling, EMT, and angiogenesis (*Figure 3F, G*). Similarly, genes with decreased DNA methylation in the C3 subtype compared with the C4 subtype were enriched for EMT and ECM regulation, whereas genes with decreased DNA methylation in the C4 subtype were significantly enriched for immune-related pathways (*Figure 3H*).

Collectively, the similar differential biological patterns of DNA methylation, gene expression, and protein levels among the CCCRC subtypes highlighted their role in influencing the TME of CRC.

## Discovery of a nongenetic tumor evolution pattern

Our study sought to investigate whether there is a dominant evolutionary pattern among the four CCCRC subtypes. According to the theory of nongenetic tumor evolution, a dominant evolutionary pattern would involve gradual changes in certain molecular features across tumor subtypes, either increasing or decreasing (*Tavernari et al., 2021*). To begin with, we obtained the common set of all DMGs (FDR <0.05), DEGs (FDR <0.05), and DEPs (p-value <0.5) between each subtype, including both upregulated and downregulated ones, by taking their intersection. Next, we separately intersected the sets of upregulated and downregulated DMGs, DEGs, and DEPs among each subtype in the common set to explore potential evolutionary patterns. From the UpSet plot (*Lex et al., 2014*), it was evident that the evolutionary pattern from C1 to C4, followed by C2, and C3 subtypes had the same sign in difference or FC and were dominant: either all positive for increasing DNA methylation/ gene expression/protein levels (red bar), or all negative for decreasing DNA methylation/gene expression/protein levels (blue bar) (*Figure 4A–C*). For instance, the top DEGs SERPINH1 and NDUFA10 showed a gradual increase and decrease in expression from C1, C4, C2 to C3 subtypes, respectively (*Figure 4D*). Meanwhile, we found the possibility of interconversion between C2 and C3 subtypes (yellow bar), between C1 and C4 subtypes (puple bar), and between C2 and C4 subtypes (gray bar) in this evolutionary pattern. Furthermore, we intersected all the positives for increasing gene expression from C1 to C4, C2, and C3 subtypes in the CRC-AFFY and CRC-RNAseq cohorts and obtained 20 CCCRC genes (*Figure 4E*, *Supplementary file 2f*), which were associated with TGF-beta signaling and infiltrating nerves. High expression of all 20 genes was significantly associated with poor DFS prognosis. To quantify the evolutionary pattern of individual CRC patients, we performed GSVA to generate CCCRC scores. To better evaluate the molecular features of the CCCRC scores, we also analyzed the correlation between the CCCRC scores and the TME-related signatures. As expected, the CCCRC scores were strongly associated with the immunosuppressive signatures, including M2 macrophages, EMT, and angiogenesis (*Figure 3—figure supplement 1H*). The CCCRC score was the highest in the C3 subtype than in the other subtypes (*Figure 4F*), and the high CCCRC score was significantly associated with shorter OS (*Figure 4G*). Overall, our analysis implied that the four CCCRC subtypes not only had their own unique biological characteristics, but also had a dominant evolutionary pattern driven by epigenetic, transcriptional, and proteomic reprogramming.

## Differences in T cell function between CCCRC subtypes

We obtained the GEP for 7766T cells from 12 patients with CRC, including four patients with the C1 subtype, one patient with the C2 subtype, two patients with the C3 subtype, and four patients with the C4 subtype (*Supplementary file 2g*). A total of five CD4+ and four CD8+ T cell clusters were identified in tumor and normal tissues, including CD8+ intraepithelial lymphocytes (CD8+ IELs), effector memory CD8+ T cells (CD8+ Tem), recently activated effector memory or effector CD8+ T cells (CD8+ Temra/Teff), exhausted CD8+ T cells (CD8+ Tex), central memory CD4+ T cells (CD4+ Tcm) and naive CD4+ T cells (CD4+ Tn), tissue-resident memory CD4+ T (CD4+ Trm) cells, TH1-like cells, Treg cells, and T cycling cells (*Figure 5—figure supplement 1A, B*). The markers of the T cell clusters were summarized in *Supplementary file 2h*. *Figure 5A and B* shows the distribution of the 10T cell clusters among each CCCRC subtype. The bulk RNAseq analyses demonstrated that C2 and C4 subtypes showed relative upregulation of immune components. Notably, we found that the C4 subtype was enriched in CD8+ Tem and CD8+ Temra/Teff cells, but lacked CD8+ Tex cells compared with the C2 subtype (*Figure 5C, D*). Within the subset of CD8+ Tex cells, we distinguished two smaller subsets according to their gene expression markers, KLRG1+ CD8+ Tex cells and HSPA1B+ CD8+ Tex cells (*Figure 5—figure supplement 1C, D*, ). KLRG1+ CD8+ Tex cells were more enriched in C1, C2, and C3 subtypes than the C4 subtype (*Figure 5E*), which resemble terminally exhausted T cells, and they were associated with non-response to ICB therapy (*Luoma et al., 2022*). Moreover, the higher ratio of KLRG1-to-CD8A expression, the worse the OS of patients in CRC-AFFY and CRC-RNAseq cohorts (*Figure 5F, G*). Meanwhile, we re-clustered the Treg cells and identified four Treg cell subsets, namely, TXNIP+ Treg cells, TNFRSF4+ Treg cells, HSPA1A+ Treg cells, and IFIT1+ Treg cells (*Figure 5—figure supplement 1E–H*). We found that TNFRSF4+ Treg cells were significantly

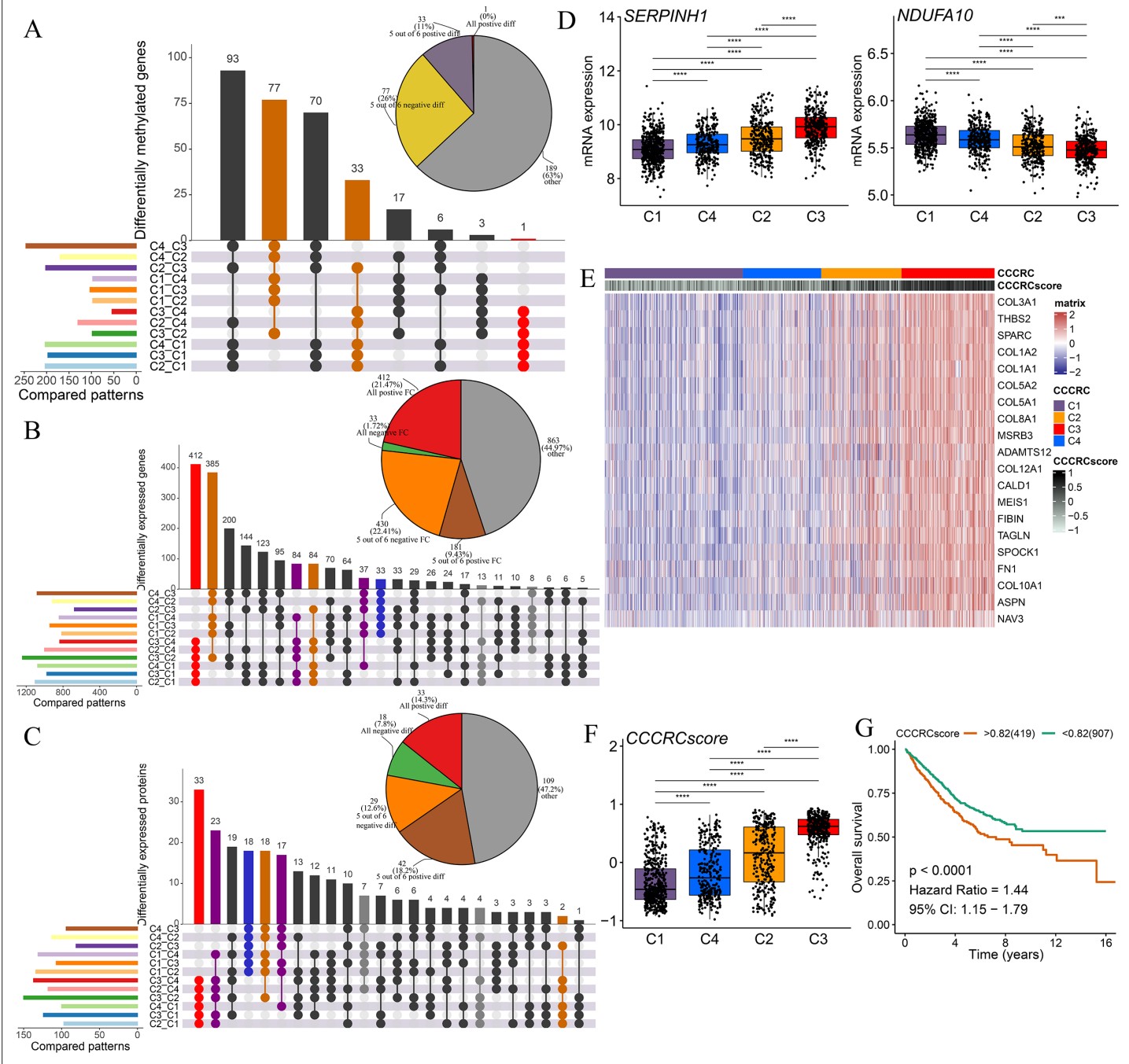

**Figure 4.** Discovery of a nongenetic tumor evolution pattern. The UpSet plots are used to visualize the intersection of differentially methylated genes (**A**), all differentially expressed genes (**B**), and all differentially expressed proteins (**C**) between each comprehensive characterization of colorectal cancer (CCCRC) subtype. As an example, 'C4_C3' means that the molecular features of the C4 subtype were upregulated compared with the C3 subtype. Pie chart (top right) distributions of the sign of differences or fold changes computed for all differentially methylated genes (**A**), differentially expressed genes (**B**), and differentially expressed proteins (**C**). (**D**) mRNA expression of two top differentially expressed genes (*y*-axis) in the CRC-AFFY cohort stratified by the CCCRC subtypes (*x*-axis). (**E**) Heatmap shows gene expression levels of 20 CCCRC genes among the four CCCRC subtypes. (**F**) Box plots show differences in the CCCRC score among the four CCCRC subtypes in the CRC-AFFY cohort. (**G**) Kaplan–Meier method of overall survival (OS) among the four CCCRC subtypes in the CRC-AFFY and CRC-RNAseq cohorts. FC: fold change; diff: difference. \*\*\*p-value <0.001; \*\*\*\*p-value <0.0001.

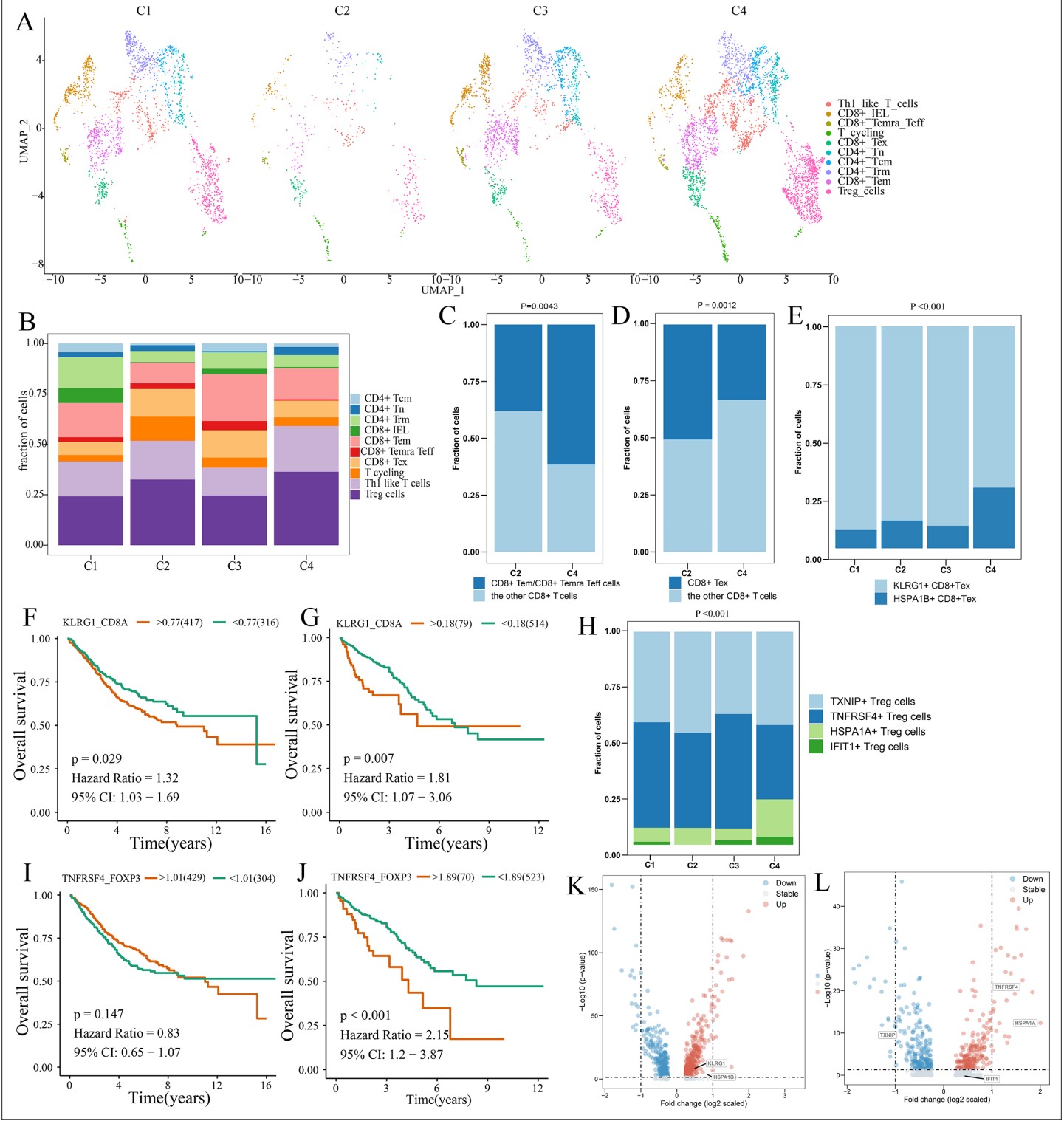

**Figure 5.** Differences in T cell function between comprehensive characterization of colorectal cancer (CCCRC) subtypes. (**A**) UMAP shows the composition of T cells colored by cluster and divided by the CCCRC subtype in colorectal cancer (CRC) tissues. (**B**) Histogram shows the cell distribution of 10T cell types in the different CCCRC subtypes. (**C**) Proportion of effector memory CD8+ T cells (CD8+ Tem), recently activated effector memory or effector CD8+ T cells (CD8+ Temra/Teff), and the other CD8+ T cells (shown in the histogram) in the C2 and C4 subtypes. (**D**) Proportion of exhausted CD8+ T cells (CD8+ Tex) and the other CD8+ T cells (shown in the histogram) in the C2 and C4 subtypes. (**E**) Histogram shows the cell distribution of KLRG1+ CD8+ Tex and HSPA1B+ CD8+Tex cells in the different CCCRC subtypes. Kaplan–Meier method with log-rank test of overall survival (OS) in the CRC-AFFY cohort (**F**) and the CRC-RNAseq cohort (**G**) between low and high ratios of KLRG1-to-CD8A expression in patients. (**H**) Histogram shows

*Figure 5 continued on next page*

Figure 5 continued

the cell distribution of TXNIP+ Treg cells, TNFRSF4+ Treg cells, HSPA1A+ Treg cells, and IFIT1+ Treg cells in the different CCCRC subtypes. Kaplan–Meier method with log-rank test of OS in the CRC-AFFY cohort (I) and the CRC-RNAseq cohort (J) between low and high ratios of TNFRSF4-to-CD8A expression in patients. (K) Volcano plot shows differentially expressed genes between tumor (red dots) and normal CD8+ T cells (blue dots). (L) Volcano plot shows differentially expressed genes between tumor (red dots) and normal Treg cells (blue dots).

The online version of this article includes the following figure supplement(s) for figure 5:

**Figure supplement 1.** Differences in T cell function between comprehensive characterization of colorectal cancer (CCCRC) subtypes.

**Figure supplement 2.** Survival analyses of T cell subsets.

more enriched in C2 and C3 subtypes than the C4 subtype (*Figure 5H*), which might indicate that TNFRSF4+ Treg cells were closely related to the formation of the tumor stroma. The higher ratio of TNFRSF4-to-FOXP3 expression, the worse the OS of patients in CRC-AFFY and CRC-RNAseq cohorts (*Figure 5I, J*). Equally important, patients with a high ratio of KLRG1-to-CD8A expression or a high ratio of TNFRSF4-to-FOXP3 expression who received anti-PD1/PDL1 therapy had a shorter OS and progression-free survival (PFS) than those with a low ratio of KLRG1-to-CD8A expression or a low ratio of TNFRSF4-to-FOXP3 expression in Gide, Hugo, Jung, and IMvigor210 datasets (*Figure 5—figure supplement 2A–H*). We also found that the expression of KLRG1 and TNFRSF4 was higher in CD8+ T cells and Treg cells, respectively, in tumor tissues than in adjacent tissues (*Figure 5K, L*). Overall, we used scRNAseq data to analyze the differences in T cell function among the different CCCRC subtypes, and the C2 subtype did show more immunosuppression than the C4 subtype, which was consistent with the bulk RNAseq analyses.

## Significance of CCCRC in guiding clinical treatment of CRC

The 5-fluorouracil (5-FU)-based chemotherapy, anti-VEGF (vascular endothelial growth factor) (bevacizumab), and anti-EGFR (epidermal growth factor receptor) (cetuximab) therapies are the first-line treatment options for CRC (*Andrei et al., 2022*). We further explored whether the different CCCRC subtypes could predict therapeutic efficacy. In the CRC-AFFY cohort, 564 patients with stage II and III CRC had chemotherapy-related clinical information, including 323 who were not treated by chemotherapy and 241 who were treated by chemotherapy. Furthermore, 155 stage II and III CRC patients with or without chemotherapy in the GSE103479 dataset were also included in our study. We found that C1 patients with stage II and III CRC receiving chemotherapy had a better OS than those who did not and were more suitable for 5-FU-based chemotherapy in the CRC-AFFY cohort and the GSE103479 dataset (*Figure 6A, B*). Furthermore, 79 mCRC patients in the GSE104645 dataset were treated with chemotherapy and 83 were treated with a combination of chemotherapy and bevacizumab. The response rate (RR) after chemotherapy (including partial response and complete response) of C1 and C4 subtypes tended to be higher than that of C2 and C3 subtypes (*Figure 6—figure supplement 1A*), whereas the RR of the C2 subtype treated with a combination of chemotherapy and bevacizumab tended to be higher than that of the other subtypes (*Figure 6—figure supplement 1B, C*). In addition, the RR tended to be higher in the C2 subtype treated with 5-FU-based chemotherapy and bevacizumab than in those treated with chemotherapy alone (*Figure 6—figure supplement 1D*). The GSE104645 dataset also contained 94 mCRC patients without the *RAS* mutation who were treated with anti-EGFR antibody. The disease control (DC) rates (DCR) after anti-EGFR therapy (including partial response, complete response, and stable disease) were 87% for C1, 64% for C2, 68% for C3, and 80% for C4, respectively (p-value = 0.23) (*Figure 6—figure supplement 1E*). The DC rate of the C1 subtype with anti-EGFR therapy tended to be higher than that of the other subtypes (p-value = 0.07) (*Figure 6—figure supplement 1F*). Notably, PFS of the C1 subtype with anti-EGFR therapy tended to be better than that of the other subtypes (log-rank p-value = 0.067) and OS of the C1 subtype was significantly better than that of the other subtypes (log-rank p-value = 0.0091) (*Figure 6C, D*). The above results suggested that the C1 subtype may benefit from chemotherapy and anti-EGFR treatment, whereas the C2 subtype may benefit from a combination of 5-FU-based chemotherapy and bevacizumab, but there was no evidence that the C3 subtype would benefit from these treatments.

To further explore the treatment strategies of the CCCRC subtypes, we trained a pre-clinical model based on a filtered gene set comprised of 335 CCCRC subtype-specific and cancer cell-intrinsic gene

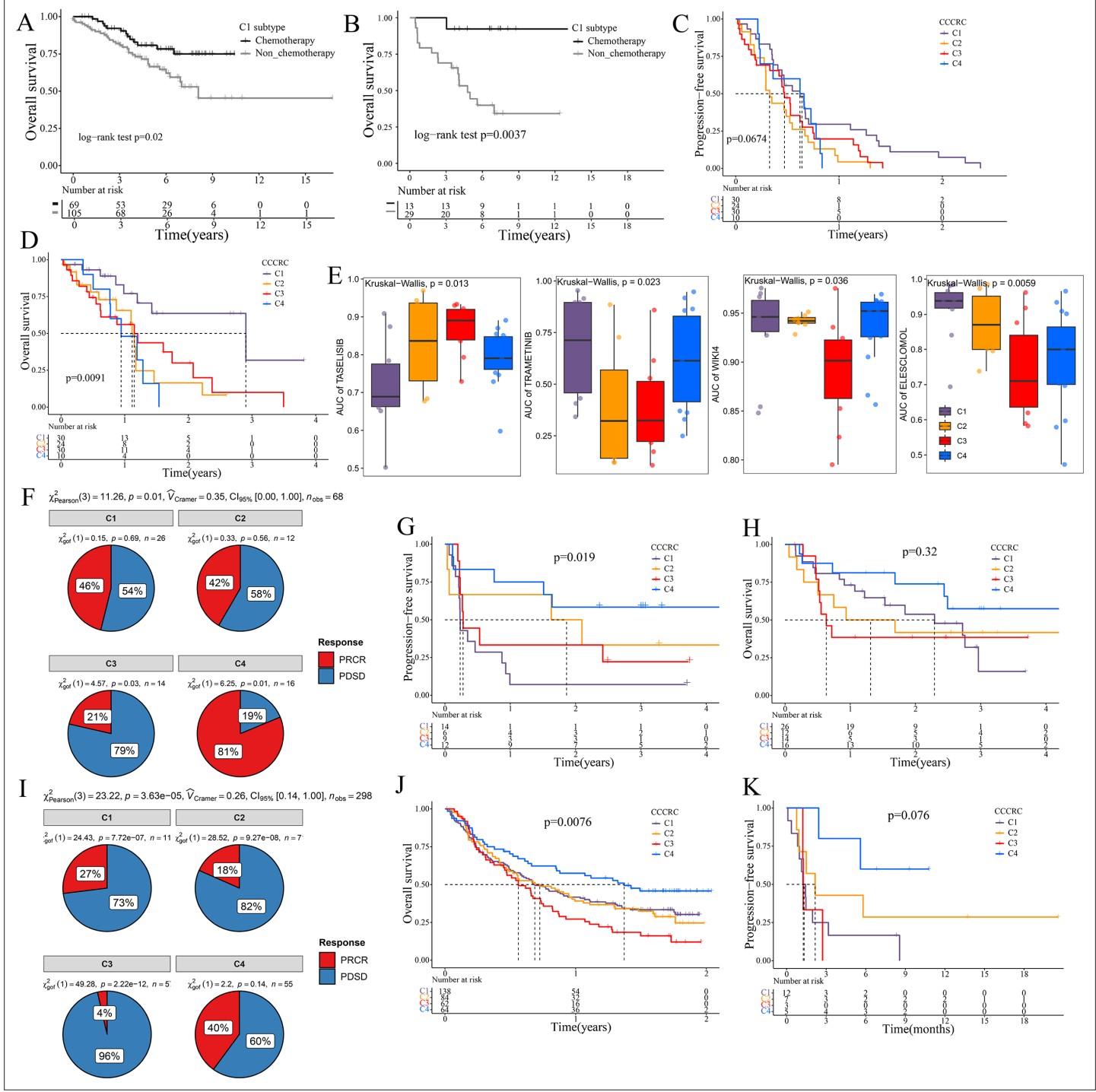

**Figure 6.** Significance of comprehensive characterization of colorectal cancer (CCCRC) in guiding the clinical treatment of colorectal cancer (CRC). Kaplan–Meier method of overall survival (OS) between stage II and III CRC C1 patients with or without chemotherapy in the CRC-AFFY cohort (**A**) and the GSE103479 (**B**) dataset. Kaplan–Meier method of progression-free survival (PFS) (**C**) and OS (**D**) among the four CCCRC subtypes in the GSE104645 dataset. (**E**) Box plots show the differences in the area under the receiver operator characteristics curve (AUC) of drug responses among the four CCCRC subtypes. (**F**) Pie chart shows the differences in the proportion of responses to immune checkpoint blockade treatment among the four CCCRC subtypes in the two independent melanoma cohorts (Gide and Hugo datasets, *n* = 68) treated with anti-PD1 therapy. Kaplan–Meier method with log-rank test of PFS (**G**) and OS (**H**) among the four CCCRC subtypes in the two independent melanoma cohorts (Gide and Hugo datasets, *n* = 68) treated with anti-PD1 therapy. (**I**) Pie chart shows the differences in the proportion of responses to immune checkpoint blockade treatment among the four CCCRC subtypes in the urothelial carcinoma cohort (*n* = 298) treated with anti-PDL1 therapy. Kaplan–Meier method with log-rank test of OS and PFS

*Figure 6 continued on next page*

*Figure 6 continued*

among the four CCCRC subtypes in the urothelial carcinoma cohort (*n* = 348) (**J**) and the lung cancer cohort (*n* = 27) (**K**) treated with anti-PD1/PDL1 therapy. PRCR: partial response and complete response; PDSD: progressive disease and stable disease.

The online version of this article includes the following figure supplement(s) for figure 6:

**Figure supplement 1.** Significance of comprehensive characterization of colorectal cancer (CCCRC) in guiding clinical treatment of colorectal cancer (CRC).

**Figure supplement 2.** Spatial transcriptomics analysis of the Cytassist sample.

**Figure supplement 3.** Spatial transcriptomics analysis of the Visium sample.

**Figure supplement 4.** Establishment of the comprehensive characterization of colorectal cancer (CCCRC) classifier.

markers (*Supplementary file 3a*). The pre-clinical model was publicly available at https://github.com/XiangkunWu/pre_clinical_model (copy archived at *Wu, 2023b*). The 71 human CRC cell lines were classified into four CCCRC subtypes (*Supplementary file 3b*). As expected, the C4 subtype cell lines demonstrated the highest TMB values and MSI frequency while exhibiting the lowest FGA scores when compared to other subtypes (*Figure 6—figure supplement 1G–I*). In contrast, C1 and C3 subtype cell lines showed significantly higher FGA scores and significantly lower TMB values and MSI frequency. The C2 subtype cell lines had median FGA scores, TMB values, and MSI frequency. Our analysis revealed that the AUCs of several pathway inhibitors differed significantly among CCCRC subtypes (*Figure 6E*, *Figure 6—figure supplement 1J*). A lower AUC value on the dose–response curve of a drug indicates an increased sensitivity of this drug to tumor treatment. Specifically, the AUCs of taselisib and GSK2110183, the PI3K pathway inhibitor, were significantly lower in the C1 subtype. The AUCs of trametinib, a MEK 1/2 inhibitor, were significantly lower in the C2 and C3 subtypes. Using GSEA in the CRC-AFFY cohort. we also found that the KEGG MAPK signaling pathway was upregulated in C2 and C3 subtypes than the other subtypes (*Figure 6—figure supplement 1K, L*). The AUCs of WIKI4, an inhibitor of the WNT pathway, elesclomol, an inducer of oxidative stress and cuproptosis, (5Z)-7-oxozeaenol, a TAK1 inhibitor, and WEHI-539, a BCL-XL inhibitor, were significantly lower in the C3 subtype. In contrast, the AUC of 5-FU was significantly higher in the C3 subtype.

ICB therapy has recently emerged as a highly promising therapeutic strategy for various malignancies, but it lacks effective markers to identify suitable patients (*Jin and Sinicrope, 2022*). We collected multiple ICB therapy-associated datasets to evaluate whether the CCCRC classification system could be used as a tool to predict ICB therapy efficacy. In the Gide and Hugo datasets treated with anti-PD1 therapy, patients were classified into the four CCCRC subtypes. As expected, the RR of anti-PD1 treatment was 81% for the C4 subtype compared to 21% for the C3 subtype (*Figure 6F*), and PFS and OS were prolonged for the C4 subtype (*Figure 6G, H*). Similar findings were observed in the IMvigor210 and Jung datasets treated with anti-PD1/PDL1. RR was significantly higher in patients with the C4 subtype (40%) compared with the other subtypes (C1 with 17%, C2 with 18%, C3 with 4%) in the IMvigor210 dataset (*Figure 6I*). The C4 subtype in the IMvigor210 and Jung datasets had the best prognosis, while patients with the C3 subtype had the worst prognosis (*Figure 6J, K*). In conclusion, we have used extensive data analysis to develop treatment protocols for each subtype, but this study is pre-clinical and still requires extensive experimentation and clinical trials for validation.

## Spatial transcriptomics analysis

We conducted a re-analysis of two CRC spatial transcriptomics (ST) data to explore the spatial distribution relationship between four CCCRC subtypes of tumor cells, T cells, and mesenchymal cells (Materials and methods). The Cytassist and Visium samples had a total of 9080 and 2660 spots, respectively. We used 'ssGSEA' method to quantify the six cell subpopulations of each spot and also visualized only the spots corresponding to the top 25% of the score ranking for each cell type (*Figure 6—figure supplement 2A, B*, *Figure 6—figure supplement 3A, B*). In Cytassist samples, we observed different spatial distribution patterns of the four subtypes of tumor cells (*Figure 6—figure supplement 2B*). Specifically, the C3 subtype of tumor cells was predominantly located in the tumor periphery with an enrichment of mesenchymal cells and T cells (areas selected by black dashed circles). In contrast, the C4 subtype of tumor cells was mainly present in the center of the tumor, accompanied by the presence of T cells. The C1 and C2 subtypes of tumor cells were distributed in relatively uniform areas, mainly in the tumor periphery, with fewer mesenchymal cells and T cells. However, the distribution areas of

C2 and C3 subtypes of tumor cells also partially were in overlap (the area selected by red dashed circles). The same distribution patterns can also be observed in the Visium sample (*Figure 6—figure supplement 3B*). Further analysis of the correlation between the ssGSEA scores of each cell type in the cell-type-rich regions and those of other cell types was conducted (*Figure 6—figure supplement 2D, E*, *Figure 6—figure supplement 3D, E*). We found that in the C3 subtype-rich region of tumor cells, the C3 subtype score of tumor cells was significantly positively correlated with the mesenchymal cell score, while in the T cell-rich region, the C3 subtype score of tumor cells was significantly negatively correlated with the T cell score. The C4 subtype score of tumor cells was significantly positively correlated with the T cell score and negatively correlated with the mesenchymal cell score in the C4 subtype-rich, T cell-rich, and mesenchymal cell-rich regions. The C1 and C2 subtype scores of tumor cells were negatively correlated with mesenchymal cell and T cell scores. Overall, these results were generally consistent with previous histopathologic analysis findings.

## Construction and validation of CCCRC classifier

To promote the widespread application of the CCCRC classification system, we developed a simple gene classifier to predict CCCRC subtypes. The CRC-AFFY cohort was randomly divided into the training and validation sets at a ratio of 7:3. We established the CCCRC classifier on the training set by utilizing multiple machine learning algorithms based on the GEP of 80 upregulated subtype-specific genes (Materials and methods, *Supplementary file 3c*). Upon application to the test set, GSE14333, and GSE17536 datasets, the performance of the eXtreme Gradient Boosting (xgboost) algorithm was the best with the highest accuracy values and *F*1 scores compared to the random forest (RF), support vector machine (SVM), and logistic regression algorithms (*Figure 6—figure supplement 4*). Notably, the CCCRC classifier based on the xgboost algorithm displayed robust performance across gene expression platforms, Affymetrix and RNA-sequencing platforms, exhibiting a balanced accuracy of >80% for all subtypes (*Supplementary file 3d*). These findings demonstrated the stability and cross-platform applicability of our classifier. The CCCRC classifier based on the xgboost algorithm is publicly available at https://github.com/XiangkunWu/CCCRC_classifier (copy archived at *Wu, 2023c*) and the CCCRC subtype information of CRC patients can be obtained by directly inputting the GEP of 80 upregulated subtype-specific genes. The CCCRC classifier might facilitate the discovery of new biomarkers and the personalized treatment of clinical patients with CRC.

## Discussion

The key role of the TME in dynamically regulating tumor progression and affecting treatment outcomes has been widely recognized, and treatment strategies targeting the TME have become a promising approach for cancer therapy (*Bejarano et al., 2021*; *Binnewies et al., 2018*; *Pandey et al., 2022*; *Tang et al., 2021*). However, there are few comprehensive analyses that consider the tumor cells and the TME as a whole. The comprehensive dissection of the crosstalk between tumor cells and TME may reveal new tumor biology concepts and identify therapeutic targets, and ultimately achieve precise medical treatment (*Bejarano et al., 2021*; *Jia et al., 2022*). Thus, we collected the molecular features of the tumor cells and TME to reconstruct the whole tumor composition and performed integrated analyses to understand the TME. The four CCCRC subtypes had distinct molecular and histopathologic characteristics, therapeutic efficacy, and prognosis (*Figure 7*). We identified a nongenetic evolutionary pattern from C1, C4, C2, and C3 was associated with an evolution from a cold (C1) to a hot (C4) and eventually suppressive (C2) and excluded (C3) microenvironment (*Figure 7*).

In this study, we identified four subtypes with distinct TME features through unsupervised clustering analysis of approximately 2000 CRC patients. C1 and C4 subtypes are typical desert and inflamed tumors, respectively, while C2 and C3 subtypes were difficult to classify into one of the classical immunophenotypes of the three-category immune classification system ('desert', 'excluded', and 'inflamed' phenotypes) (*Chen and Mellman, 2017*; *Rosenberg et al., 2016*) based on TME features due to the unclear distribution of stromal components and lymphocytes. Our pathologists evaluated the histopathologic characteristics for each subtype under the microscope and observed that the C2 subtype was mainly categorized as an excluded phenotype and the C3 subtype was mainly classified as a desert and an excluded phenotype. However, the WSIs showed that lymphocytes in the C2 subtype were more frequently intermixed with the stroma within but not adjacent to the main

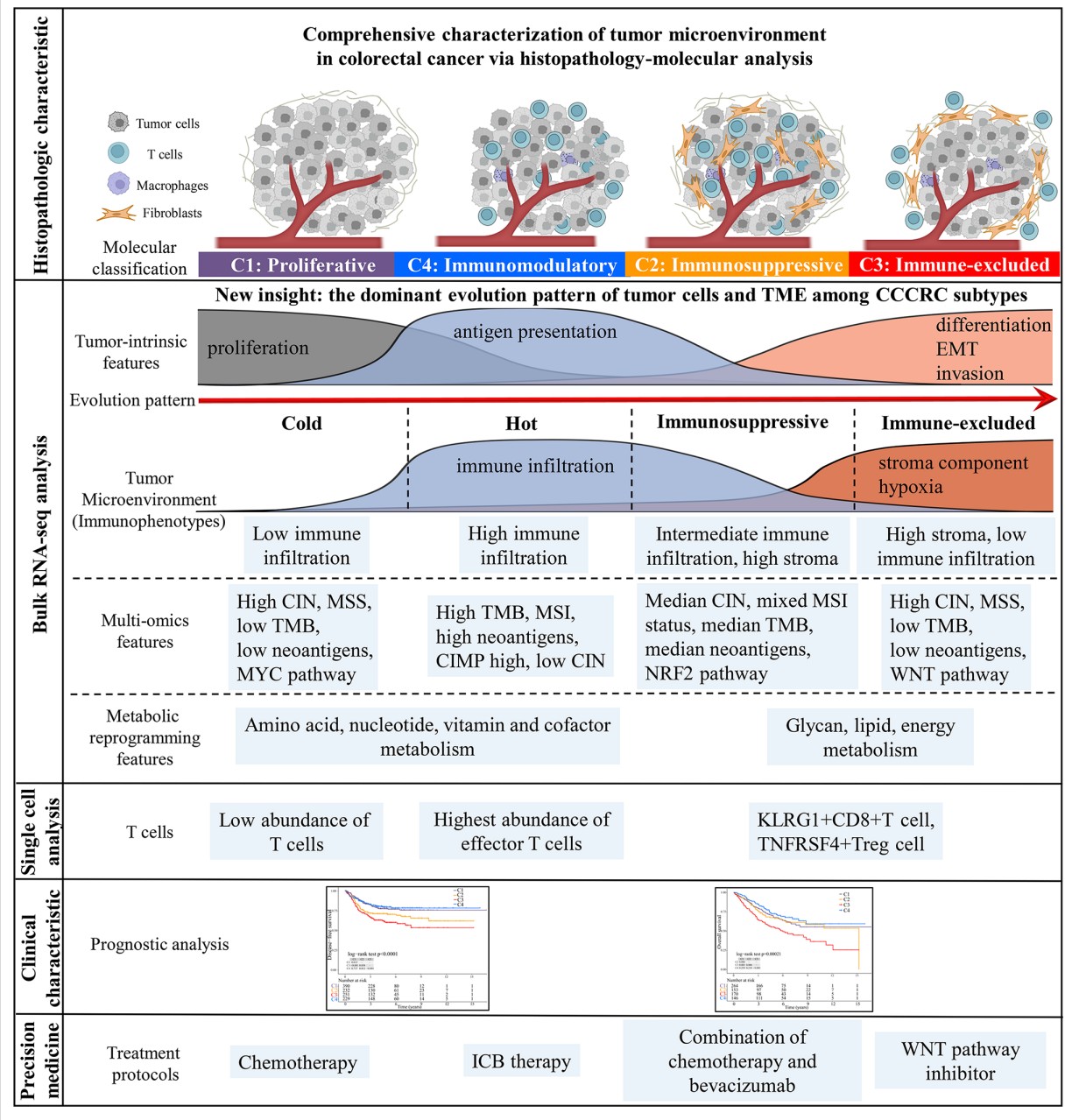

**Figure 7.** Overview of characteristics of comprehensive characterization of colorectal cancer (CCCRC) subtypes. The graphical abstract summarizes histopathologic characteristics, tumor microenvironment features, multi-omics features, scRNAseq features, treatment strategies, and prognostic value for CCCRC subtypes.

tumor mass, and lymphocytes in the C3 subtype were more frequently excluded from the tumor mass but not intermixed with lymphocytes within the main tumor mass, both of which were classified as the excluded phenotype. Notably, we used AI-enabled spatial analysis of WSIs to confirm the semi-quantitative results of the pathologist, that is, the C2 subtype had increased lymphocyte and stromal infiltration in CT and IM regions and the C3 subtype had the highest abundance of stroma and less lymphocyte infiltration in the CT region, while lymphocyte infiltration and stromal components were observed in the IM region, which were more visually confirmed in the results of ST data analysis. We also found that the C2 subtype had the highest T cell dysfunction score and the C3 subtype had the highest T cell exclusion score. GSEA demonstrated that the terminally exhausted CD8+ T cell signature was upregulated in the C2 subtype compared with the C4 subtype. scRNA-seq analysis

showed that KLRG1+ CD8+ T cells were significantly more enriched in C2 and C3 subtypes than the C4 subtype. KLRG1+ CD8+ T cells were associated with nonresponse to ICB therapy, which were more terminally differentiated than KLRG1− CD8+ T cells and had lower proliferative capacity (*Luoma et al., 2022*). KLRG1 is a marker of terminal differentiation of CD8+ T cells (*Luoma et al., 2022*), and the inhibitory receptor of ILC1s (group 1 innate lymphoid cells), ILC2s, and NK cells (*Chiossone et al., 2018*). ILC1s in tumors express high levels of the KLRG1 gene and pro-angiogenic activity and may even promote tumor progression in TGF-beta-rich tumors (*Chiossone et al., 2018*). Therefore, we defined C2 and C3 subtypes as immunosuppressed and immune-excluded, respectively. Our CCCRC classification system refined the three-category immune classification system (*Chen and Mellman, 2017*; *Rosenberg et al., 2016*). Moreover, we defined for the first time the four-category immune classification system based on the integrative multi-omics analysis and histopathologic characteristics ('hot', 'immunosuppressed', 'excluded', and 'cold' phenotypes) (*Galon and Bruni, 2019*; *Kirchhammer et al., 2022*).

We observed an intriguing nongenetic evolution pattern among the CCCRC subtypes, wherein a shift occurred from the C1 (proliferative subtype) to C4 (immunomodulatory subtype) subtype, followed by the C2 (immunosuppressed subtype) subtype and finally, the C3 (immune-excluded subtype) subtype. We proposed that in this evolutionary pattern, immune infiltration gradually increased with the augmentation of genomic alterations and tumor immunogenicity, while the stromal components also increased in a gradual manner. These components are crucial in the progression of tumor, as they contribute to the exhaustion and eventual exclusion of lymphocytes from the tumor bed (*Calon et al., 2015*; *Galon and Bruni, 2019*; *Kirchhammer et al., 2022*). Meanwhile, our research revealed that both C2 and C3 subtypes exhibited a high level of tumor stroma, while C1 and C4 subtypes were characterized by active DNA damage and repair and high tumor proliferation. Additionally, C2 and C4 subtypes had an abundance of immune components. This was consistent with our finding that there may be interconversion between the C1 and C4 subtypes, between the C4 and C2 subtypes, and between the C2 and C3 subtypes in this evolutionary pattern. The interconversion between C2 and C4 subtypes in this evolutionary pattern was the rarest situation, indicating that once the tumor enters the C2 subtype, it was difficult to reverse and will progress to the C3 subtype. Not incidentally, this evolutionary pattern was consistent with the findings of *Tavernari et al., 2021*, who demonstrated that the progression from lepidic to solid histology of lung adenocarcinoma was associated with a transition from a cold (lepidic) to a hot (papillary and especially acinar), and eventually suppressive and excluded (solid) TME. Using spatially resolved molecular profiles, they uncovered the coexistence of cold, inflamed, and excluded regions within individual tumors. Similarly, our ST analysis revealed the simultaneous presence of multiple CCCRC subtypes in the same sample. What is more, we identified a gene list that promoted this evolutionary pattern and proposed CCCRC score based on the gene list to quantify the evolutionary pattern of individual CRC patients, which served as an independent prognostic predictor. However, it is important to note that further experimental evidence is required to verify this bold speculation of the evolutionary pattern, and a large collective effort is needed to arrive at a consensus.

The CCCRC subtypes were significantly correlated with the CMS subtypes (*Guinney et al., 2015*) and clinicopathological characteristics. The CMS classification system integrated six independent classification systems utilizing a network-based approach (*Guinney et al., 2015*), which is considered as the most robust classification system that is used to predict prognoses and to guide ICB therapy, chemotherapy, and anti-EGFR therapy as well as to screen new potential targeted drugs (*Eide et al., 2017*; *Kawazoe et al., 2020*; *Linnekamp et al., 2018*; *Sveen et al., 2018*; *Thota et al., 2021*; *Zhan et al., 2021*). In our study, the transcriptomic data used to identify CCCRC subtypes were essentially the same as the cohort used to develop the CMS subtypes (*Guinney et al., 2015*). The C1, C2/C3, and C4 subtypes exhibited partial overlap with the CMS2, CMS4, and CMS1 subtypes, respectively. Notably, we observed a higher prevalence of right-sided lesions in the C4 subtype, consistent with the CMS1 subtype, and a higher prevalence of left-sided lesions in the C1/C3 subtype, consistent with the CMS4 subtype. Previous studies indicated that left-sided CRCs had greater genetic instability and right-sided CRCs were associated with MSI and CIMP-high features (*Bufill, 1990*; *Lee et al., 2017*). We observed that patients with the CMS1 subtype, characterized by high immune infiltration and activation, did not have the best prognosis compared with the other CMS subtypes, while patients with the CMS2 subtype, characterized by low immune infiltration, had the best prognosis (*Becht et al., 2016a*;

*Guinney et al., 2015*). This might be due to the fact that CMS1 subtype had a higher accumulation of fibroblasts compared to CMS2 subtype (*Guinney et al., 2015*). Moreover, our study revealed that, in comparison with the C4 subtype, the CMS1 subtype had fewer anti-tumor immune components, more stroma components, and other immunosuppressive components, resulting in a poorer prognosis. The prognosis of patients with the C2 subtype was also better than that of patients with the CMS4 and C3 subtypes, and the C2 subtype within the CMS4 subtype also had a better prognosis than the C3 subtype within the CMS4 subtype, which might be related to the presence of more immune infiltration in the C2 subtype than in the C3 and CMS4 subtypes. Our CCCRC classification system further subdivided CMS4 into two subtypes, the C2 and C3 subtypes, each with distinct molecular and prognostic characteristics. Meanwhile, the C4 subtype with MSI exhibited higher immune infiltration and less stromal component compared with the CMS4 subtype with MSI. Overall, our CCCRC classification system can be combined with the CMS classification system and other molecular subtypes to enhance the understanding of the association between TME heterogeneity and different clinical phenotypes.

The CCCRC classification system hold promise for discovering novel biomarkers and therapeutic targets in the TME based on one or more immune evasion mechanisms, which can be further validated by combining TIDE score (*Jiang et al., 2018*), CMS classification system (*Guinney et al., 2015*) and forthcoming molecular classification systems. The C1, C2, and C3 subtypes corresponded to immune evasion mechanisms of low immunogenicity, T cell dysfunction, and T cell exclusion, respectively. In contrast, the C4 subtype exhibited the highest degree of infiltration by anti-tumor immune components and the most favorable prognosis, potentially indicating immunological control of the tumor, known as immune equilibrium. For instance, KDM1A and RAD21 are specific genes for C1 subtype, and it has been reported that interfering with them could activate interferon signaling and induce tumor intrinsic immunogenicity (*Deng et al., 2022*; *Nguyen et al., 2022*; *Supplementary file 2d*). Similarly, SPP1, a feature gene for the C3 subtype, has been reported to be positively correlated with colon cancer liver metastasis in SPP1+ macrophages and highly expressed in CMS4 (mesenchymal subtype) (*Lee et al., 2020*; *Liu et al., 2022*). SPP1+ macrophages interacted with CAFs to induce extracellular matrix remodeling and the formation of an immune-excluded phenotype, thereby limiting immune infiltration into the tumor core (*Liu et al., 2023*). SERPINB9 is significantly upregulated in the C2 subtype, and previous research has demonstrated its pivotal function in enhancing tumor cell survival and promoting the existence of immune-suppressive cells within the TME (*Jiang et al., 2020*). In this study, the expression levels of SPP1 and SERPINB9 were found to be significantly positively correlated with the exclusion and dysfunction scores (TIDE score), respectively (*Supplementary file 2e*). Meanwhile, we identified a large number of mutant genes significantly enriched in the C4 subtype, which mutated to cause substantial immune infiltration and could be candidate genes for mRNA vaccine development. The RNA-mediated immunotherapy regulating the TME is known as the next era of cancer treatment (*Pandey et al., 2022*). Additionally, the CCCRC classification system has the potential to offer a deeper understanding of the underlying biology of CRC by integrating forthcoming molecular classification systems. First, it is necessary to identify the commonalities and differences between different classification systems and how they can complement each other. This can be achieved by conducting comparative studies and analyzing the overlap between different classification systems. Second, it is essential to leverage the strengths of each classification system to identify new biomarkers or therapeutic targets. For example, the CCCRC classification can provide valuable insights into the underlying immune evasion mechanisms of CRC. Third, it is important to combine the discovered potential therapeutic targets with different classification systems for mutual verification. Overall, extensive research has demonstrated a significant correlation between the CCCRC subtype-specific genes and immune evasion mechanisms, underscoring their potential as promising therapeutic targets for developing precision medicine approaches against cancer within the TME. However, realizing the full potential of these classification systems will require a collaborative effort among researchers and clinicians to ensure their effective integration and utilization. Through integration, we can leverage the strengths of different classification systems to identify new biomarkers and therapeutic targets that can ultimately improve patient outcomes.

The CCCRC classification system may contribute to the design of future clinical trials. In our study, we found that the C1 subtype may be suitable for chemotherapy and cetuximab treatment, which is consistent with the predicted results of CMS2 in the CALGB/SWOG 80405l clinical trial (*Lenz et al., 2019*). Additionally, we observed that the C2 subtype could potentially benefit from a combination

of chemotherapy and bevacizumab treatment. By analyzing the molecular characteristics of the C4 subtype and validating it in multiple cohorts receiving ICB therapy, we proposed that ICB therapy may represent a promising therapeutic strategy for the treatment of the C4 subtype. However, our predicted results showed that the C3 subtype was resistant to chemotherapy, cetuximab, bevacizumab, and ICB therapy. Therefore, we used cell line drug sensitivity assay data to explore further treatment options for the CCCRC subtypes. We observed that the C1 subtype exhibited higher PI3K pathway activity compared to the other subtypes. As expected, our drug sensitivity analyses suggested that the C1 subtype may benefit from PI3K inhibitors. Notably, the potential pro-inflammatory effect of PI3K-mTOR pathway inhibitors might offer advantages for cancer immunotherapy (*Fruman et al., 2017*), indicating that the C1 subtype might be suitable for PI3K inhibitors combined with ICB treatment. Similarly, we also found upregulation of the KEGG MAPK signaling pathway in the C2 and C3 subtypes. Drug sensitivity analyses suggested that the C2 and C3 subtypes may benefit from treatment with trametinib, an MEK 1/2 inhibitor. *Borriello et al., 2017* demonstrated that activation of CAFs in primary human neuroblastoma depends on co-activation of JAK2/STAT3 and MEK/ERK1/2 signaling in tumor cells, and trametinib can enhance the response of tumor cells to etoposide. Moreover, *Datta et al., 2022* showed that combined inhibition of MEK and STAT3 can alleviate stromal inflammation, reduce the abundance of iCAF, myCAF, and apCAF, and remodel the TME to overcome immune therapy resistance in pancreatic ductal adenocarcinoma. We also found that the C3 subtype showed increased sensitivity to the WNT pathway inhibitor WIKI4. The C3 subtypes had an upregulation of the WNT signaling pathway, and activation of the WNT signaling pathway can cause T cells in highly immunogenic tumors to be excluded from the tumor core (*Takeuchi et al., 2021*). Overall, our classification system can assist in identifying populations suitable for appropriate treatments in a clinical setting and may promote precision medicine, but this research is pre-clinical and requires further substantial basic experiments and clinical trials to validate.

The CCCRC classification system also can facilitate the understanding of the differences in the results of the CALGB/SWOG 80405 and FIRE-3 clinical trials (*Heinemann et al., 2014*; *Lenz et al., 2019*). Both trials evaluated the combination of cetuximab or bevacizumab with a different chemotherapy backbone: in the CALGB/SWOG 80405 trial, 75% of patients received oxaliplatin, while in the FIRE-3 trial, all patients received irinotecan. *Aderka et al., 2019* discussed the reasons for the differences in the results of these two clinical trials, which may be related to differences in the chemotherapy backbone used and TME heterogeneity. Based on our examination of the results summarized in *Figure 4* of the work by *Aderka et al., 2019*, we found that differences in the treatment outcomes of the CMS1 and CMS4 subtypes were the crucial factor behind the divergent results observed in the two clinical trials. The CMS1 and CMS4 subtypes had a microenvironment rich in CAFs. Our CCCRC classification results also showed that CMS1, in addition to mainly consisting of the C4 subtype, also contained a considerable number of the C2 subtype, while the CMS4 subtype mainly consisted of the C2 and C3 subtypes. Furthermore, our study results indicated that the C2 subtype was suitable for chemotherapy in combination with bevacizumab, possibly because the combination can inhibit the CAFs and abnormal blood vessel formation in the TME, thus alleviating the immune suppression of the immune cells. However, the C3 subtype was not suitable for chemotherapy in combination with bevacizumab because it only accumulated CAFs and abnormal blood vessel formation but lacked T cell infiltration. Therefore, we boldly speculated that the CMS1 and CMS4 subtypes in the CALGB/SWOG 80405 clinical trial may contain more C2 subtypes than those in the FIRE-3 clinical trial, leading to the CMS1 and CMS4 subtypes in the CALGB/SWOG 80405 clinical trial being more suitable for chemotherapy in combination with bevacizumab than cetuximab compared to the FIRE-3 clinical trial. Overall, the integration of CCCRC and CMS classification systems can provide valuable insights for understanding the divergent outcomes of the two clinical trials.

Our study has several limitations that should be considered when interpreting the results. First, the present study is a pre-clinical research and further validation of our findings requires additional basic experiments and prospective clinical trials. Second, it should be noted that due to the difficulty in evaluating the impact of removing individual TME-related signatures on clustering results, we performed cluster sensitivity analysis by removing the entire TME category. Third, given the relatively small sample size of scRNAseq and ST data used in this study, larger datasets are needed for analysis to better understand the complexity of the TME.

To conclude, our study proposed the CCCRC classification system and performed integrated data analysis to clearly characterize the molecular features and histopathologic characteristics of each CCCRC subtype, develop the corresponding personalized treatments for patients with the different CCCRC subtypes, and construct the simple gene classifier to facilitate clinical application. We believe that our study will serve as a research paradigm for dissecting the TME and for transitioning from molecular classification to clinical translation, thereby accelerating the understanding of the TME in CRC and contributing to the development of therapeutic targets against TME.

## Materials and methods

### Acquisition and processing of GEP for the investigation of CCCRC

A total of 2196 samples were obtained from ten publicly available datasets (*Supplementary file 1c*). The eight publicly available raw microarray datasets sequenced by the Affymetrix gene chip were downloaded from the Gene Expression Omnibus (GEO; https://www.ncbi.nlm.nih.gov/geo/) and renormalized by the robust multi-array average (RMA) method, including GSE13067, GSE13294, GSE14333, GSE17536, GSE33113, GSE37892, GSE38832, and GSE39582. Samples that overlapped in GSE14333 and GSE17536 datasets were excluded from the GSE14333 dataset. The two RNA sequencing datasets (TCGA and CPTAC datasets) were obtained from the TCGA data portal (March 2022) (https://portal.gdc.cancer.gov/), and the count data were normalized by the 'voom' method. Ensembl IDs were annotated into gene symbols using GENCODE (v36). If the gene symbol had multiple probes or duplicates, the median value was calculated as its relative GEP. Before merging the microarray or RNAseq datasets into the CRC-AFFY or CRC-RNAseq cohort, the batch effects were examined using PCA and corrected using the 'Combat' function. The selection criteria of these patients included: (1) CRC primary tissue samples; (2) coming from the same sequencing platform; (3) surgically resected specimens. The exclusion criteria included: (1) CRC metastatic tissue; (2) puncture tissues; (3) having received radiation therapy or chemotherapy prior to surgery. Detailed information on the sample size and the corresponding clinicopathological data of the CRC-AFFY and CRC-RNAseq cohorts are summarized in *Supplementary file 1c* and *Supplementary file 1f*. The CPTAC dataset was excluded from the survival analysis due to a median follow-up time of less than 3 years. Additionally, samples with a follow-up time of zero were also excluded from the follow-up analysis.

### Calculation of the TME-related signature scores

After reviewing previously published studies, the Molecular Signatures Database (MSigDB; http://www.gsea-msigdb.org/gsea/msigdb/index.jsp), and the Reactome pathway portal (https://reactome.org/PathwayBrowser/), we identified relevant biomarker genes for tumor, immune, stromal, and metabolic reprogramming signatures. The 4525 genes from each of the 61 TME-related signatures are listed in *Supplementary file 1a*, as well as the source of each signature. GSVA with default parameters using the R package 'GSVA' was performed to calculate the signature score of each TME-related signature for each sample of each cohort separately based on the relative GEP (*Hänzelmann et al., 2013*).

### Normal tissue versus tumor tissue analysis

To assess the distribution of normal and tumor samples in the GSE39582 (*n* = 19 normal) and TCGA (*n* = 41 normal) datasets, the GEP of each dataset were re-normalized, including the normal samples (consistent with the description of data normalization above). PCOA based on 'Euclidean' distance was used to analyze the distribution between normal and CRC samples (*Dixon, 2003*). PERMANOVA test (*Zhu et al., 2021*) was used to evaluate whether the difference in 'Euclidean' distances between the normal and CRC samples was statistically significant (obtained using the R package 'vegan'; *Desgarennes et al., 2014*).

### Comprehensive characterization of CRC

The 'ConsensusClusterPlus' function of the R package 'ConsensusClusterPlus' (*Monti et al., 2003*) was applied to identify the optimal number of CCCRC based on the TME-related signatures in the CRC-AFFY cohort (partitioning around medoids (pam) clustering; 'Pearson' distance; 1000 iterations; from 2 to 6 clusters). The stability of the clusters was evaluated using the consensus matrices heatmap,

the empirical CDF plot, and the delta area plot. The purpose of the consensus matrices heatmap was to find the 'cleanest' cluster partition, with dark blue color block indicating that samples were often clustered together with a high consensus and white color block indicating that samples were not often clustered together with a low consensus. CDF plot displayed the consensus distribution for each *k* and the delta area plot represented the change in the area under the CDF curves, and both were used to determine the maximum consensus distribution. To verify the repeatability and robustness of the CCCRC subtypes in the CRC-RNAseq cohort, we used the 'pamr.predict' function of the R package 'pamr' (*Tibshirani et al., 2002*) to extract centroids of each subtype and establish a PAMR classifier in the CRC-AFFY cohort. A threshold with minimum 10-fold cross-validation error was selected to identify the TME-related signatures that exhibit at least one non-zero difference between each subtype (seed = 11). PAM is a statistical technique to identify subsets of features that best characterize each class using nearest shrunken centroids (*Tibshirani et al., 2002*). The technique is general and can be used in many other classification problems. The TME-related signature scores were normalized by the Z-scores before performing 'pamr.predict' analysis. PCOA based on 'Euclidean' distance was used to analyze the distribution of the CCCRC subtypes.

## Estimation of the TME cell abundance with other methods

The cell abundance of each sample was estimated based on the GEP using the MCP-counter algorithm (*Becht et al., 2016b*) and the CIBERSORT (*Newman et al., 2015*) algorithm, both of which have been validated using the GEP of the corresponding cell populations and the degree of cellular infiltration estimated by immunohistochemistry. The MCP-counter algorithm estimated the cell abundance of nine immune and stromal cell populations. The CIBERSORT algorithm, which applies the LM22 matrix, estimated the cell fraction of 22 immune cell populations. The ESTIMATE algorithm with default parameters was utilized to estimate the degree of infiltration of the total immune cells and stromal cells in the TME of each sample, as well as the tumor purity (*Yoshihara et al., 2013*).

## Calculation of the other biological pathway enrichment scores

Human metabolism-related pathways were obtained from the KEGG database (https://www.genome.jp/kegg/). The 1660 genes assigned to 86 human metabolism-related pathways are listed in *Supplementary file 3e*. Ten oncogenic signatures containing 331 genes and the terminally exhausted T cell signature were retrieved from a previously published study (*Beltra et al., 2020*; *Sanchez-Vega et al., 2018*; *Supplementary file 3e*). GSVA was performed to calculate the enrichment score of each signature for each sample of each cohort separately based on the relative GEP. To identify the potential differences in the biological functions of genes among CCCRC subtypes, GSEA was performed based on the gene signatures using the R package 'clusterprofiler' (*Yu et al., 2012*).

## Histopathologic examination of the TCGA-CRC samples

A total of 616 TCGA CRC diagnostic HE-stained WSIs were downloaded from the TCGA data portal (March 2022) (https://portal.gdc.cancer.gov/), and the WSIs were examined blindly by two experienced pathologists. A total of 254 WSIs were included after removing the WSIs with poor quality and without views of the IM (*Supplementary file 1i*). According to the semi-quantitative pathological assessment of lymphocytes and their spatial location with malignant epithelial cells, the pathologist classified CRC into three immunophenotypes: 'desert', 'excluded', and 'inflamed', as previously described (*Chen and Mellman, 2017*; *Rosenberg et al., 2016*). The inflamed phenotype was characterized by abundant lymphocytes in direct contact with malignant cells, the excluded phenotype was characterized by lymphocytes merely present in the stroma within or adjacent to the main tumor mass, and the desert phenotype was characterized by the lack of lymphocytes and stroma.

## Artificial intelligence-enabled spatial analysis of CRC-WSIs

We performed artificial intelligence (AI)-enabled spatial analysis of WSIs and developed a CRC-multiclass model to identify eight tissue types: tumor, stroma, lymphocyte, normal colon mucosa, debris, adipose, mucin, and muscle, and quantified the abundances of the tumor, stroma, and lymphocytes in the CT region and the IM region, respectively. The CRC-multiclass model consisted of two sequential parts: a muscle/non-muscle classifier that could distinguish each muscle (MUS) patch in HE-stained WSIs, and a seven-tissue classifier that could classify seven tissue types: adipose

(ADI), debris (DEB), lymphocytes (LYM), mucus (MUC), normal colon mucosa (NORM), cancer-associated stroma (STR), and colorectal adenocarcinoma epithelium (TUM). To develop the CRC-multiclass model, we used three dataset including NCT-CRC-HE-100k dataset, NCT-CRC-HE-7k dataset (https://zenodo.org/record/1214456#.YyRJGWB6RmM), and TCGA-CRC dataset. At first, pathologists conducted quality control on the NCT-CRC-HE-100K dataset. Tissues with inaccurate labeling were removed from the dataset. The stromal tissue dataset was reduced from 10,446 patches to 9638 patches, and the lymphocyte tissue dataset was reduced from 11,555 patches to 10,453 patches. After up-sampling stroma tissue patch, 4000 additional patches were added. Total of 90,686 patches were selected. The dataset contains patches that are 224 pixels by 224 pixels in size and was generated from 85 patients. Then, 9 WSIs from 9 patients in the TCGA-CRC dataset were annotated with tissue label, resulting in a total of 4288 patches. The preprocessing step involved identifying the outline of the entire tissue and then segmenting it. Segmentation was performed by non-overlapping 224 × 224 patch extraction based on the tissue outline, and a patch was considered valid if 75% of its area fell within the annotation boundaries. We did not do any modification on NCT-CRC-7k dataset, which includes 7180 patches from 50 patients with size of 224 pixels by 224 pixels. To develop the muscle/non-muscle model, The NCT-CRC-HE-100k and NCT-CRC-HE-7k datasets were merged and subsequently partitioned into an internal training set of 68,506 patches, an internal validation set of 14,679 patches, and an internal test set of 14,681 patches. This model predicts whether a tissue patch is a non-muscle patch. If the probability of a sample belonging to the non-muscle class is greater than 0.99, it is classified as non-muscle and then would be fed to our seven-tissue classifier. The seven-tissue classifier was trained with 54,597 patches and validated with 23,400 patches on the CRC-HE-100k dataset that underwent quality control and removal of muscle tissue, and the muscle-free CRC-HE-7k dataset of 5741 patches was used as the internal test set. The 4288 patches of TCGA-CRC dataset are the external testing set for evaluating externally for these two models.

The WSI tissue type prediction pipeline was as follows. First, the background was removed by the preprocessing steps. The preprocessing step for WSI tissue segmentation involved converting the color space from RGB to HSV, applying a median filter to the saturation channel to remove noise, and finally using a thresholding approach to segment the WSI tissue from background. Second, the tissues regions of WSIs were segmented into non-overlapping image patches at a resolution of 0.5 μm/pixel (×20 magnification). It is worth noting that if the WSI consisted of ×40 magnification, it was downsampled to ×20 magnification. Next, the image patches were fed into the CRC-multiclass model. If an image patch was determined to be non-muscle by the muscle/non-muscle classifier, it was fed into the seven-tissue classifier to predict its tissue class. Both the muscle/non-muscle classifier and the seven-tissue classifier used the same model architecture, which involved utilizing a pre-trained ResNet50 model for transfer learning. The last layer of the feature extraction layer was connected to two fully connected layers, with ReLU as the activation function and a dropout of 0.4 applied to the first fully connected layer. The final activation function used was softmax. During training, the optimizer used was stochastic gradient descent, with a learning rate of 0.001 and momentum of 0.9. The muscle/non-muscle classifier was trained for 20 epochs, while the seven-tissue classifier was trained for 10 epochs. During this experiment, we tested three model architectures, including ResNet50, vgg16, and Inception V3 for the CRC-tissue classifier. According to the accuracy of seven tissues (tumor, stroma, lymphocytes, normal colon mucosa, debris, adipose, and mucin) in the NCT-CRC-HE-7k dataset, the performance of ResNet50 was the best, which was the reason we selected ResNet50 as the basic model architecture.

After recognizing the CRC tissue types by our deep learning model automatically, we quantified the abundances of the tumor, lymphocytes, and stroma in the CT region and the IM region. The quantification pipeline consisted of three steps. First, we used the open-source software QuPath-0.3.2 (https://qupath.github.io/; *Bankhead et al., 2017*) to delineate the CT and IM region. The IM region was defined as 500 mm outside the CT region (*Kather et al., 2018*). The CT and IM regions were manually annotated by two experienced pathologists to reduce bias. Next, we utilized our CRC-multiclass model to predict the tissue type of patches within the CT and IM regions after segmenting images within the annotated CT and IM contours into 224-pixel by 224-pixel patches. This enabled us to determine the number of lymphocyte (LYM), stroma (STR), and tumor (TUM) patches within these regions. Then, the abundances of the tumor, lymphocytes, and stroma in different regions of each WSI

were quantified with an area ratio of their area to region area. A total of 254 TCGA-CRC WSIs were quantified with this quantification pipeline.

## Acquisition of signatures associated with the ICB therapy response

The TIDE score was calculated using GEP, and it was used to evaluate the degree of T cell dysfunction and T cell exclusion (*Jiang et al., 2018*). The higher the score, the later the dysfunction stage of T cells or the higher the degree of T cell exclusion. The gene expression average of all samples in each cohort was used as the normalized control and the normalized GEP was uploaded to the TIDE website (http://tide.dfci.harvard.edu/).

## Acquisition and processing of CRC multi-omics data

Masked somatic mutation data (*n* = 571 samples), masked copy number segment data (*n* = 609 samples), and DNA methylation beta values (Illumina human methylation 450) (45 normal samples and 390 tumor samples) were download from the TCGA data portal (March 2022) (https://portal.gdc.cancer.gov/). The liquid chromatography–tandem mass spectrometry (LC–MS/MS)-based proteomic data for the TCGA CRC samples (*n* = 88 samples) were obtained from a previously published study (*Zhang et al., 2014*). The R package 'maftools 2.6.05' with default parameters was used to analyze the somatic mutation data. Synonymous mutations were regarded as wild-type, and genes with mutation rates <5% were excluded. Nonsynonymous mutations were used to calculate tumor mutation burden (TMB). SCNAs defined by the GISTIC2.0 module on the GenePattern website (https://www.genepattern.org/), including arm-level gain (1), and high amplification (2), diploid/normal (0), arm-level deletion (−1), and deep deletion (−2). The CINmetrics algorithm was used to calculate CIN signature, including SCNAs count and FGA, which was proposed by Vishaloza et al. (https://rdrr.io/github/lasseignelab/CINmetrics/; *Oza et al., 2023*) based on previously published studies (*Baumbusch et al., 2013*; *Chin et al., 2007*; *Davison et al., 2014*). If somatic mutation events or SCNAs occurred in one or more genes in the oncogenic pathway, the tumor sample was considered altered in a given pathway. The MSI status was obtained from the CMS website (https://www.synapse.org/#!Synapse:syn2623706). Tumor neoantigen signature and the ITH data in the CRC-RNAseq cohort were obtained from a previously published study (*Thorsson et al., 2018*). The prevalence of somatic mutation events or SCNAs was compared among CCCRC cases using Fisher's exact test or chi-square test. For the DNA methylation data, probes located in promoter CpG islands were extracted, including TSS200, 1stExon, TSS1500, and 5′UTR. The probes detected on X and Y chromosomes or any probe with NA value were removed. For genes with multiple probes mapped to the promoter, the median beta value was calculated as the degree of gene methylation. The beta-value difference was defined as the difference between the mean beta value of each CCCRC sample and normal samples, and Wilcoxon rank-sum test was used to test whether the difference was statistically significant. DMGs between normal and CRC samples were defined as |mean beta value| < 0.2 in normal samples and |mean beta value| > 0.5 in CRC samples with FDR <0.05.

We performed differential analysis between each of the four subtypes and the remaining subtypes using Wilcoxon rank-sum test to identify DMGs with FDR <0.001 and DEPs with p-value <0.05. The difference of DMGs and DEPs was defined as the difference between each of the four subtypes and the remaining subtypes. The 'limma' package was used to identify DEGs with FDR <0.001 between each of the four subtypes and the remaining subtypes. To identify subtype-specific DMGs, DEPs, and DEGs in one of the subtypes, we excluded those that were found to be differentially expressed in comparisons between one of the other subtypes and the remaining subtypes. Additionally, we performed differential analysis of methylation gene, gene expression, and protein levels between each subtype based on the above method. Enrichment analysis was performed by the R package 'clusterProfiler' (*Yu et al., 2012*). p-values were adjusted for multiple comparisons by the Benjamini–Hochberg-corrected FDR.

## Regulon analysis

The R package 'RTN' was used to reconstruct the transcriptional regulatory networks of regulons (*Robertson et al., 2017*), including 31 transcription factors and 82 chromatin remodeling genes, that were associated with CRC (*Vymetalkova et al., 2019*; *Xu et al., 2021*; *Supplementary file 3f*). Mutual information and Spearman's correlation analysis were utilized to infer the possible associations

between a regulator and all possible targets from the GEP, and the permutation algorithm was used to eliminate associations with an FDR $>1 \times 10^{-5}$. Unstable associations were removed by bootstrap analysis ($n = 1000$), and the weakest association in triangles consisting of two regulators and common targets were eliminated by the data processing inequality algorithm. Two-tailed gene set enrichment analysis was used to calculate the regulon activity score for each sample.

## Publicly available CRC classification systems

To classify CRC samples into different CRC subtypes according to the previously published gene classifier, gene lists for the four classifiers were extracted from relevant publications and summarized (*Supplementary file 3g*). These classifiers included the Budinska classification system (*Budinska et al., 2013*), the De Sousa classification system (*De Sousa E Melo et al., 2013*), the Roepman classification system (*Roepman et al., 2014*), and the Sadanandam classification system (*Sadanandam et al., 2013*). The nearest template prediction (NTP) algorithm was employed to classify the samples and to generate an FDR to assess the classification robustness. For NTP implementation, we screened genes that were specifically and positively associated with one subtype according to the screening strategies of a previously published study (*Medico et al., 2015*).

## Bulk RNAseq and scRNAseq data processing of the GSE108989 dataset

A total of 12 CRC samples with bulk RNAseq and scRNAseq data were obtained from the GSE108989 dataset (*Zhang et al., 2018*; *Supplementary file 2g*). To identify the CCCRC subtypes, bulk RNAseq with transcripts per million (TPM) was further log2-transformed, and GSVA was performed to calculate the signature score of each TME-related signature in each sample based on the GEP. The PAMR classifier was used to classify CRC samples into four CCCRC subtypes based on the TME-related signatures (seed = 11, threshold = 0.566). For scRNAseq data processing, the raw GEP were normalized and selected according to the following criteria: cells with >200 genes and <7000 genes and <20% of mitochondrial gene expression in UMI counts, which was determined using the R package 'Seurat'. Counts of the filtered matrix for each gene were normalized to the total library size with the Seurat 'NormalizeData' function. The 'FindVariableGenes' function was used to identify 2000 hypervariable genes for unsupervised clustering. Next, each integrated feature was centered to a mean of zero and scaled by the standard deviation with the Seurat 'ScaleData' function. The 'RunPCA' function was used for PCA. We identified diverse T cell clusters using the 'FindClusters' function, and set the resolution parameter to 0.5. Each cell cluster was compared to the other clusters by the 'FindAllMarkers' function to identify DEGs (only pos: TRUE, min.PCt: 0.25, logFc.threshold: 0.25). Cell annotation was carried out by consulting the latest cell marker databases, such as CellMarker (https://www.biolegend.com/en-us/cell-markers) and PanglaoDB (https://ngdc.cncb.ac.cn/databasecommons/database/id/6917), combined with a previously published study (*Zhang et al., 2018*). To define the feature genes for each CCCRC subtype, differential expression analysis between CCCRC subtypes was performed using the 'FindMarkers' function. FDR <0.05 were considered statistically significant.

## Collection and processing of therapy-associated datasets

Therapy-associated datasets were used to explore the treatment strategies for each CCCRC subtype. GEP of GSE103479 and GSE104645 datasets were downloaded from the GEO database. If the gene symbol was annotated with multiple probes, the median value was used as the expression of the gene. The clinical data of the GSE104645 dataset were obtained from the supplementary file of a study by *Okita et al., 2018*. The GSE103479 dataset contained 156 stage II and III CRC patients with or without 5-FU-based chemotherapy. The GSE104645 dataset contained 193 mCRC patients treated with chemotherapy, a combination of chemotherapy and bevacizumab, or anti-EGFR therapies. The available RNAseq expression dataset of patients treated with anti-PD-1 therapy was also downloaded. The Gide (PRJEB23709) dataset was downloaded, and the raw fastq files was re-analyzed. The RNA reads were aligned using STAR v2.5.3 and quantified as TPM using RSEM v1.3.0 and log2-transformed. Ensembl IDs were annotated into gene symbols using GENCODE v36. The GEP of Hugo (GSE78220) and Jung (GSE135222) datasets and the corresponding clinical data were downloaded from the GEO database, and the FPKM values were converted to log2-transformed TPM values. We obtained the GEP ($n = 348$) of urothelial carcinoma patients in the IMvigor210 dataset treated with

anti-PD-L1 therapy and the corresponding clinical data using the R package 'IMvigor210CoreBiologies 1.0.0' (IMvigor210 dataset), and the count values were converted to log2-transformed TPM values. To reduce batch effects and tissue type-specific effects, we first performed GSVA analysis of the TME-related signatures in each dataset, and the signature scores were normalized by Z-scores method. Next, we used the PAMR classifier to classify the samples into the four CCCRC subtypes based on the TME-related signatures in each dataset (seed = 11, threshold = 0.566). Detailed information on the sample size and the corresponding treatment data of the therapy-associated datasets are summarized in *Supplementary file 3h*.

## Construction of a pre-clinical model

To explore the treatment for each CCCRC subtype using cancer cell line drug sensitivity experiments, we developed a pre-clinical model based on subtype-specific, cancer cell-intrinsic gene markers according to a previously published study (*Eide et al., 2017*). Firstly, the 'limma' package was used to identify DEGs with FDR <0.05 between each of the four subtypes and the remaining subtypes in the CRC-AFFY cohort. To identify subtype-specific genes in one of the subtypes, we excluded those that were found to be differentially expressed in comparisons between one of the other subtypes and the remaining subtypes. The upregulated subtype-specific genes (log2FC >0 and FDR <0.05) were ranked based on their log2FC and selected the top 500 genes for further gene screening. Secondly, the GEP of human CRC tissues versus patient-derived xenografts (PDX) in the GSE35144 dataset by the R package 'limma' was used to remove those genes associated with stromal and immune components. DEGs with FDR >0.5 and log2FC <1 between human CRC tissues versus PDX were considered as cancer cell-intrinsic genes. Thirdly, we also utilized human CRC cell lines to obtained cancer cell-intrinsic genes. A total of 71 human CRC cell lines with RNAseq data (log2TPM) was obtained from the Genomics of Drug Sensitivity in Cancer (GDSC) database (https://depmap.org/portal/download/all/), 43 of which had dose–response curve (AUC) values. The MSI status, FGA and TMB information of CRC cell lines was obtained from cbioportal website (https://www.cbioportal.org/study/summary?id=ccle_broad_2019). RNAseq data for 71 human CRC cell lines were used to further determine the cancer cell-intrinsic genes and genes among the top 25% within (1) the 10–90% percentile range of the largest expression values and (2) the highest expression in at least three samples. The subtype-specific genes and cancer cell-intrinsic genes were intersected to generate the gene list for developing the pre-clinical model. The pre-clinical model was developed using the NTP function of R package 'CMScaller', which can be applied to cross-tissues and cross-platform predictions (*Hoshida, 2010*). The GEP (log2TPM) of 71 human CRC cell lines normalized by the Z-score were input into the pre-clinical model, and the cell lines were divided into four CCCRC subtypes. We then compared the drug responses among the CCCRC subtypes by analyzing the differences in the AUC values.

## Spatial transcriptomics data processing

To investigate the spatial distribution relationship between four CCCRC subtypes of tumor cells, T cells, and stromal cells, we conducted a re-analysis of publicly available CRC ST data obtained from the 10X website (https://www.10xgenomics.com/resources/datasets). The Space Ranger output files were then processed with Seurat (V4.1.1) (*Hao et al., 2021*) using SCTransform for normalization (*Hafemeister and Satija, 2019*). RunPCA were used to dimension reduction and RunUMAP to visualize the data. We used 'ssGSEA' method implemented in the R package 'GSVA' to score the six cell types (C1–C4 subtype cancer cells, mesenchymal cells, and T cells) (*Hänzelmann et al., 2013*). The 'ssGSEA' method has been previously demonstrated to be highly reliable and suitable for ST data analysis (*Wu et al., 2022*). The cell-type-rich region was defined as the ssGSEA score of each cell type from one spot larger than the 75% quantile of this cell type. The markers for the six cell types are listed in the *Supplementary file 1a* and *Supplementary file 3a*.

## Discovery and validation of the CCCRC classifier

In order to facilitate the widespread application of CCCRC classification system, we established a simple gene classifier to predict CCCRC subtypes. Firstly, we filtered genes based on their mean expression and variance in the CRC-AFFY cohort, and genes with expression and variance below the bottom 25% were removed. Then, we applied the RF algorithm in the R package 'caret' to perform feature selection on the CCCRC subtype-specific genes of the CRC-AFFY cohort. The top

20 most informative features for each subtype were ranked and selected based on the impurity measure generated by the algorithm. This allowed us to identify critical genes that are strongly associated with each CCCRC subtype and develop the CCCRC classifier. Next, we randomly divided the CRC-AFFY cohort into training and test sets at a ratio of 7:3 using 'createDataPartition' function provided in the R package 'caret' (seed = 123). The GEP was normalized with $Z$-scores prior to model training and validation. The CCCRC classifiers were trained with the top 80 upregulated subtype-specific genes using the RF, SVM, xgboost, and Logistic Regression algorithms implemented in the R package 'caret'. Finally, we validated the CCCRC classifier on the GSE14333 and GSE17536 datasets, as well as the CRC-AFFY cohort. The consensus clustering result of the CRC-AFFY cohort and the PAMR classifier result of the CRC-RNAseq cohort were used as 'gold-standard' ($n$ = 2196 samples). We evaluated the predictive performance of the CCCRC classifier by evaluating measures such as accuracy value and $F$1 score, which were generated using the 'confusionMatrix' function provided in the R package 'caret'.

## Statistical analyses

All statistical analyses were conducted by R 4.0.2 software. Statistical significance of the comparisons for continuous variables and categorical variables was assessed by the Wilcoxon rank-sum test or Kruskal–Wallis test and Fisher's exact test or chi-square test, respectively. Correlations between variables were estimated by Spearman's correlation analysis or Pearson's correlation analysis. Patients were divided into either high or low gene expression groups by the best cutoff calculated by the R package 'survminer'. The Kaplan–Meier method with log-rank test was utilized to generate the survival curves. Univariate and multivariate Cox proportional hazard regression analyses were performed to generate 95% confidence intervals and hazard ratios. Two-sided p-values <0.05 were considered statistically significant.

## Acknowledgements

We thank TCGA and GEO databases and the cBioportal website for free use. We thank Nanjing Simcere Medical Laboratory Science Co, Ltd and Jiangsu Simcere Diagnostics Co, Ltd, and all the members of its AI and bioinformatics team, for generously sharing their experience and codes.

## Additional information

### Competing interests

Xiaoping Qu, Yiran Zheng, Minghui Ge, Linlin Yan: are affiliated with Nanjing Simcere Medical Laboratory Science Co., Ltd. and Jiangsu Simcere Diagnostics Co., Ltd. These authors have no financial interests to declare. The other authors declare that no competing interests exist.

### Funding

| Funder | Grant reference number | Author |
|---|---|---|
| National Key R&D Program of China | No. 2021YFF1201004 | Li Liang |

The funders had no role in study design, data collection, and interpretation, or the decision to submit the work for publication.

### Author contributions

Xiangkun Wu, Conceptualization, Data curation, Validation, Investigation, Methodology, Writing – original draft; Hong Yan, Conceptualization, Validation, Investigation, Methodology; Mingxing Qiu, Conceptualization, Data curation, Validation, Methodology; Xiaoping Qu, Resources, Data curation, Formal analysis, Investigation, Methodology, Writing – original draft; Jing Wang, Conceptualization, Data curation, Methodology; Shaowan Xu, Yiran Zheng, Minghui Ge, Linlin Yan, Conceptualization, Investigation, Methodology; Li Liang, Conceptualization, Supervision, Funding acquisition, Investigation, Writing – review and editing

Author ORCIDs
Li Liang http://orcid.org/0000-0003-2424-4926

Decision letter and Author response
Decision letter https://doi.org/10.7554/eLife.86032.sa1
Author response https://doi.org/10.7554/eLife.86032.sa2

## Additional files

### Supplementary files

• Supplementary file 1. Identification and validation of comprehensive characterization of colorectal cancer (CCCRC) classification. (a) The 61 tumor microenvironment (TME)-related signatures used to comprehensively characterize CRC. (b) Overlapping genes between 61 TME-related signatures. (c) The datasets used to identify and validate of CCCRC classification. (d) Univariate cox analysis and Kaplan–Meier method showing the ability of the 61 TME-related signatures to predict disease-free survival and overall survival in the CRC-AFFY cohort. (e) Difference analysis of metabolism signature scores between the four CCCRC subtypes in the CRC-AFFY cohort. (f) The information of clinicopathological characteristics and CCCRC subtypes in the CRC-AFFY and CRC-RNAseq cohorts. (g) Associations between clinicopathological characteristics and CCCRC subtypes. (h) Multivariate Cox proportional hazard regression analysis for disease-free survival and overall survival adjusting for the CCCRC subtypes and clinicopathological characteristics in the combined cohort (the CRC-AFFY and CRC-RNAseq cohorts). (1) A total of 254 hematoxylin and eosin (HE)-stained whole-slide images used to evaluate histological characteristics of CCCRC subtypes.

• Supplementary file 2. The multi-omics analysis of the comprehensive characterization of colorectal cancer (CCCRC) subtypes. (a) The different frequency of somatic mutations between the four CCCRC subtypes in TCGA-CRC dataset. The different frequency of somatic copy number alterations between the four CCCRC subtypes in TCGA-CRC dataset. (b) The fraction of altered samples per oncogenic pathway in each CCCRC subtype. (c) Lists of differentially expressed genes between each of the four subtypes and the remaining subtypes and between each subtype. (d) The correlation between all genes and the exclusion and dysfunction scores obtained from the Tumor Immune Dysfunction and Exclusion (TIDE) website. (e) Univariate cox analysis and Kaplan–Meier method showing the ability of 20 CCCRC genes to predict disease-free survival in the CRC-AFFY cohort. (f) A total of 7766T cells from 12 CRC patients in GSE108989 dataset. (g) The marker genes of the T cell clusters.

• Supplementary file 3. Clinical application of comprehensive characterization of colorectal cancer (CCCRC) classification system. (a) The 335 CCCRC subtype-specific and cancer cell-intrinsic gene markers used to construct the pre-clinical model. (b) The 71 human colorectal cancer cell lines grouped according to CCCRC classification system. (c) A total of 80 upregulated subtype-specific genes used to construct the CCCRC classifier. (d) The performance of the CCCRC classifier in the CRC-AFFY and the CRC-RNAseq cohort. Lists of 86 human metabolism-related signatures, 10 oncogenic signatures and terminally exhausted T cell signature. (e) Transcription factors and chromatin modifiers in colorectal cancer. (f) Publicly available CRC molecular classification systems. (g) The datasets used to reveal the therapy strategies for four CCCRC subtypes.

• MDAR checklist

### Data availability

The datasets presented in this study can be found in online repositories. The names of the repository/repositories and accession number(s) can be found in the article. This paper does not report original codes. The CCCRC classifier is publicly available at https://github.com/XiangkunWu/CCCRC_classifier (copy archived at *Wu, 2023c*). All software is freely or commercially available and is listed in the Methods. Any additional information required to reanalyze the data reported in this work paper is available from the lead contact upon request.

The following previously published datasets were used:

| Author(s) | Year | Dataset title | Dataset URL | Database and Identifier |
|---|---|---|---|---|
| Jorissen RN | 2009 | Expression data from primary colorectal cancers | https://www.ncbi.nlm.nih.gov/geo/query/acc.cgi?acc=GSE13067 | NCBI Gene Expression Omnibus, GSE13067 |
| Jorissen RN | 2009 | Expression data from primary colorectal cancers | https://www.ncbi.nlm.nih.gov/geo/query/acc.cgi?acc=GSE13294 | NCBI Gene Expression Omnibus, GSE13294 |
| Jorissen RN | 2010 | Expression data from 290 primary colorectal cancers | https://www.ncbi.nlm.nih.gov/geo/query/acc.cgi?acc=GSE14333 | NCBI Gene Expression Omnibus, GSE14333 |
| Smith JJ | 2009 | Metastasis Gene Expression Profile Predicts Recurrence and Death in Colon Cancer Patients (Moffitt Samples) | https://www.ncbi.nlm.nih.gov/geo/query/acc.cgi?acc=GSE17536 | NCBI Gene Expression Omnibus, GSE17536 |
| Medema JP | 2011 | AMC colon cancer AJCCII | https://www.ncbi.nlm.nih.gov/geo/query/acc.cgi?acc=GSE33113 | NCBI Gene Expression Omnibus, GSE33113 |
| Laibe S | 2012 | A seven-gene signature aggregates a subgroup of stage II colon cancers with stage III | https://www.ncbi.nlm.nih.gov/geo/query/acc.cgi?acc=GSE37892 | NCBI Gene Expression Omnibus, GSE37892 |
| Tripathi MK | 2014 | NFAT transcriptional activity is associated with metastatic capacity in colon cancer | https://www.ncbi.nlm.nih.gov/geo/query/acc.cgi?acc=GSE38832 | NCBI Gene Expression Omnibus, GSE38832 |
| Marisa L | 2013 | Gene expression Classification of Colon Cancer defines six molecular subtypes with distinct clinical, molecular and survival characteristics [Expression] | https://www.ncbi.nlm.nih.gov/geo/query/acc.cgi?acc=GSE39582 | NCBI Gene Expression Omnibus, GSE39582 |
| Allen WL | 2017 | Gene expression data from stage II and III treated and untreated colorectal cancer patients | https://www.ncbi.nlm.nih.gov/geo/query/acc.cgi?acc=GSE103479 | NCBI Gene Expression Omnibus, GSE103479 |
| Okita A | 2018 | Consensus molecular subtypes classification of colorectal cancer as a predictive factor for chemotherapeutic efficacy against metastatic colorectal cancer | https://www.ncbi.nlm.nih.gov/geo/query/acc.cgi?acc=GSE104645 | NCBI Gene Expression Omnibus, GSE104645 |
| Gide TN et al | 2019 | Biomarkers of response and resistance to checkpoint blockade immunotherapy in metastatic melanoma | https://www.ncbi.nlm.nih.gov/bioproject/PRJEB23709 | NCBI BioProject, PRJEB23709 |

*Continued on next page*

*Continued*

| Author(s) | Year | Dataset title | Dataset URL | Database and Identifier |
|---|---|---|---|---|
| Hugo W, Zaretsky JM, Sun L, Song C, Homet-Moreno B, Hu-Lieskovan S, Berent-Maoz B, Pang J, Chmielowski B, Cherry G, Seja E, Lomeli S, Kong X, Kelley MC, Sosman JA, Johnson DB, Riba A | 2016 | mRNA expressions in pre-treatment melanomas undergoing anti-PD-1 checkpoint inhibition therapy | https://www.ncbi.nlm.nih.gov/geo/query/acc.cgi?acc=GSE78220 | NCBI Gene Expression Omnibus, GSE78220 |
| Jung H, Kim J | 2019 | DNA methylation loss coupled with mitotic cell division promotes immune evasion of tumours with high mutation load [RNA-seq] | https://www.ncbi.nlm.nih.gov/geo/query/acc.cgi?acc=GSE135222 | NCBI Gene Expression Omnibus, GSE135222 |
| Nickles D, Senbabaoglu Y, Sheinson D | 2018 | TGF-b attenuates tumor response to PD-L1 blockade by contributing to exclusion of T cells | http://research-pub.gene.com/IMvigor210CoreBiologies/#transcriptome-wide-gene-expression-data | IMvigor210, 10.1038/nature25501 |
| 10x GENOMICS | 2023 | Human Colorectal Cancer, 11 mm Capture Area (FFPE) | https://www.10xgenomics.com/resources/datasets/human-colorectal-cancer-11-mm-capture-area-ffpe-2-standard | 10x GENOMICS, /human-colorectal-cancer-11-mm-capture-area-ffpe-2-standard |
| 10x GENOMICS | 2020 | Human Intestine Cancer (FPPE) | https://www.10xgenomics.com/resources/datasets/human-intestine-cancer-1-standard | 10x GENOMICS, /human-intestine-cancer-1-standard |
| Zhang L | 2018 | Lineage tracking reveals dynamic relationships of T cells in colorectal cancer | https://www.ncbi.nlm.nih.gov/geo/query/acc.cgi?acc=GSE108989 | NCBI Gene Expression Omnibus, GSE108989 |

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
