## [Editor Report]

This study represents a valuable body of work in which the authors assemble a molecular description of colorectal cancer and a classification into subtypes. Overall, the evidence supporting the findings is solid, and consensus over a diverse range of data from publicly available sources is convincing. When added to existing knowledge this work may contribute to future biomarker discoveries for colorectal cancer.

---

## [Decision Letter]

**Decision letter after peer review:**

Thank you for submitting your article "Comprehensive characterization of tumor microenvironment in colorectal cancer via histopathology-molecular analysis" for consideration by *eLife*. Your article has been reviewed by 3 peer reviewers, one of whom is a member of our Board of Reviewing Editors, and the evaluation has been overseen by Tony Ng as the Senior Editor.

Essential revisions:

We consider it essential that the following points are addressed in the revision, but also please consider all recommendations from the reviewers below, and refer to them for further details on these essential points:

1) Please comment on and show evidence for the robustness of the derived 4 classes (C1-C4).

2) Comment on the degree of intra-patient heterogeneity of CCCRC subtypes.

3) Please be explicit about why gene sets were chosen, and their completeness in the context of CRC, and any overlap between them.

4) Clarify why clinicopathological covariates were not included.

5) Clarify the derivation of the single-sample gene classifier, specifying exactly the genes considered and in the final classifier. Please make sure the final classifier is available.

6) To support the main claim that the CCCRC classification can facilitate biomarker discovery, please comment on how best to combine the CCCRC classification together with existing (and future) classifications, to achieve this aim.

*Reviewer #1 (Recommendations for the authors):*

Thank you for your manuscript. Here are my specific suggestions.

In Figure S1 A and B, better labels are required. Both seem to be labelled as TCGA. There are open circles that are not labelled.

Lines 142-144 – Sentence needs expansion. The "prognostic ability" of the signatures was evaluated. The "signatures associated with stromal and tumor compartments significantly correlated".

Line 152 has the only use of the term "TME panel scores", we presume you mean the signature scores?

Line 242 contains the first mention of the TCGA-CRC cohort, please reference the supplementary methods.

Line 642, please use "20X magnification" and "40X magnification", not "20 magnification" or "magnifications".

Line 349, "dominate" should be "dominant"?

Line 1134, I believe this is an UpSet plot, not a Venn plot. (Alexander Lex, Nils Gehlenborg, Hendrik Strobelt, Romain Vuillemot, Hanspeter Pfister. UpSet: Visualization of Intersecting Sets IEEE Transactions on Visualization and Computer Graphics (InfoVis), 20(12): 1983--1992, doi:10.1109/TVCG.2014.2346248, 2014.)

Figure 6C, D PFS/OS legend is the wrong way around.

Line 429, Please clarify where the filtered list of 81 genes came from.

Lines 452-462, Describe a single-sample gene classifier based on the CRC-RNAseq data with training and a hold-out set. It is also not clear if the 80 gene set was input to the model training or if the final gene list was.

Line 458: "The gene classifier based on the xgboost algorithm is publicly available at https://github.com/XiangkunWu/CCCRC" but the link does not contain the classifier nor the final list of genes. It has the code used but not the classifier itself. Please upload the classifier in a form that others can use.

*Reviewer #2 (Recommendations for the authors):*

– How robust is the clustering to the choice of clustering method and the choice of gene lists? For example, could the authors comment on what would happen if one of the "tumour cell" signatures is left out, given that "Energy" appears to be a relatively strong component of C2 and C3 for both CRC-AFFY and CRC-RNAseq?

This is important to expand on, at least as a Discussion point, because the CCCRC clustering (Figure 1) is a foundational part of this work, and the rest of this paper is structured around these definitions.

– Given that the 61 gene signatures appear to clearly separate CCR-AFFY and CRC-RNAseq into 4 clusters using the presented method (Figure 1A), could the authors comment on why the CCCRC and CMS subtypes were not more similar to each other (Figure 1G)? Is there a component captured by (or driving) CCCRC which is not captured by CMS, and vice versa?

– In addition to the summary in Fig1G, it would be helpful if Figure 1A heatmap was also annotated with CMS and MSI annotations, reordering the samples within each CCCRC cluster if necessary.

– The completeness (or otherwise) of the gene lists chosen for the gene panel should be made more explicit. To what extent is the chosen gene panel likely to represent the components of CRC tumour-associated immune cells, tumour cells, etc? Does the panel of gene signatures capture all the biological differences previously described between CMS subtypes, e.g. TGFb activation; WNT and MYC activation; others?

– What is the overlap in samples between CRC-AFFY and those used for the development of the CMS classification, and/or the original papers which fed into the CMS? Are all of these cohorts similar to each other, or overlapping? Was there a similar balance of subtypes and clinicopathological variables? This would help the reader to understand all possible sources of the difference between CCCRC and the current consensus CMS.

– Please clarify why clinicopathological covariates were not included in the OS and PFS models (Figures1, 4-6). For example, left-sidedness, KRAS or BRAF mutations, MSI; e.g. see Guinney (2015) Table S13.

– Fig1I: "Forest plot of multivariate Cox proportional hazard regression analysis of PFS after adjusting for TNM stage and CMS subtype in the CRC-AFFY cohort." – is this intended to mean that CMS was a covariate? If yes, why?

– The discussion comparing the reported predictive or prognostic value of CCCRC subtypes (this work) and CMS subtypes (published) could be expanded. For which endpoints did the CMS subtypes (published) have superior or similar separation than the CCCRC classification (this work), and does this affect the biological interpretation of each classification scheme?

– The claims "the C4 subtype was suitable for ICB treatment" and "significance of CCCRC in guiding the clinical treatment of colorectal cancer", and any similar phrases about guiding treatment, need to be rephrased; this is a pre-clinical study.

– To support the main claim that the CCCRC classification can facilitate biomarker discovery, can the authors comment on how best to combine the CCCRC classification together with existing (and future) classifications, to achieve this aim?

– Could expand the Discussion on whether/how this "holistic" CCCRC classification might contribute towards understanding the heterogeneity in treatment outcomes seen in different CRC trials (e.g. FIRE-3, COIN, CALGB/SWOG 80405).

*Reviewer #3 (Recommendations for the authors):*

The authors' title "histopathology-molecular analysis" suggests the manuscript includes an emphasis on the investigation of histopathological parameters. Whilst the authors do examine the relationship between histopathological analysis of specimens with CCCRC subtypes this features as a minor aspect of the study which largely focuses on large-scale investigation of transcriptional signatures.

Figures1C, 2C, 3B and 4E – pvalues on plots illegible

Methodology for tumour evolution patterns needs comment/expansion – I could not identify this in the current methods.

Comment on the degree of intra-patient heterogeneity of CCCRC subtypes would be nice. A significant degree of overlap between immunosuppressive and immune excluded, and proliferative and immuno-modulatory signatures in Figure 1A is apparent and should be commented upon.

The addition of a spatially resolved RNA seq dataset (likely available from Nanostring of 10X genomics Visium platforms) would be nice to see the distribution of CCCRC subtypes within each patient.

[Editors' note: further revisions were suggested prior to acceptance, as described below.]

Thank you for resubmitting your work entitled "Comprehensive characterization of tumor microenvironment in colorectal cancer via histopathology-molecular analysis" for further consideration by *eLife*. Your revised article has been evaluated by Tony Ng (Senior Editor) and a Reviewing Editor.

The manuscript has been improved but there are some remaining issues that need to be addressed, as outlined below:

1) Please remove the word "histopathology" from the title since that analysis was not a major driver of the results.

2) The limitations of the study have been added as requested. Could you please add a Limitations paragraph in the Discussion, to summarise all the study's limitations in one place.

3) Please add to the limitations paragraph that the removal of single signatures at the clustering stage was not evaluated. From the Response (page 13): "Since the effect of removing one of the TME-related signatures on the clustering results was not well evaluated, we attempted to remove the entire category."

4) Statements about clinical decisions still remain, for example:

Line 753: "The CCCRC classification system may facilitate clinical treatment decisions."

Line 757: "We demonstrated the suitability of C4 for immune checkpoint therapy".

and in other places.

Given the study limitations, all such statements should be toned down.

e.g. "in future … may contribute to.. subject to further work.. in combination with other classification systems.."

---

## [Author Response]

Essential revisions:We consider it essential that the following points are addressed in the revision, but also please consider all recommendations from the reviewers below, and refer to them for further details on these essential points:1) Please comment on and show evidence for the robustness of the derived 4 classes (C1-C4).

Thank you for your valuable comments on our manuscript. We have added more method details for evaluating the optimal number of clusters (Lines 149-153, 830-836, 1600-1604). In this study, consensus clustering analysis was performed based on the TME-related signature scores in the CRC-AFFY cohort, and the optimal cluster number was determined to be four using the consensus matrices heatmap, the empirical cumulative distribution function (CDF) plot, and δ area plot (Figure 1—figure supplement 2a-c). The purpose of the consensus matrices heatmap was to find the "cleanest" cluster partition, with dark blue color block indicating that samples were often clustered together with a high consensus and white color block indicating that samples were not often clustered together with a low consensus. CDF plot displayed the consensus distribution for each k and the δ area plot represented the change in the area under the CDF curves, and both were used to determine the maximum consensus distribution. At k=4, the consensus matrices heatmap displayed a distinct distribution of items with high intra-cluster consensus and low inter-cluster consensus. The CDF plot showed that k=4 yields the maximum consensus distribution. The Δ area plot showed that k=4 is the most significant k with a noticeable increase in consensus. (Lines 1600-1604)

Furthermore, we externally validated the reproducibility of the CCCRC subtypes in the CRC-RNAseq cohort, which revealed the same four CCCRC subtypes with similar patterns of differences in the TME components (Figure 1—figure supplement 2fg). PCOA also showed that the four CCCRC subtypes were distinctly separated and the P-values for intercomparisons of the euclidean distances between them were all <0.05 using PERMANOVA test in the CRC-AFFY and CRC-RNAseq cohorts (Figure 1—figure supplement 2h). (Lines 149-153)

2) Comment on the degree of intra-patient heterogeneity of CCCRC subtypes.

We are grateful for the valuable suggestion regarding the CCCRC classification system. We have added intra-tumor heterogeneity analysis for each subtype (Lines 196-198). The level of intratumor heterogeneity (ITH) was significantly linked to poor prognosis and drug resistance (Caswell and Swanton, 2017). The ITH data used in our study for the CRC-RNAseq cohort was obtained from a previous study conducted by Thorsson et al. (Thorsson et al., 2018). As expected, the ITH of the C2 and C3 subtypes was higher than that of the other subtypes, while the ITH of the C4 subtype was the lowest (Figure 1F). Our analysis using the Kaplan-Meier method demonstrated that patients with the C4 subtype had significantly higher overall survival (OS) and disease-free survival (DFS) compared to those with the C2 and C3 subtypes. Furthermore, the C3 subtype was resistant to chemotherapy, cetuximab, bevacizumab, and ICB therapy. Our investigation of drug sensitivity data of cell lines also indicated that the C2 and C3 subtypes were generally not responsive to most drugs.

3) Please be explicit about why gene sets were chosen, and their completeness in the context of CRC, and any overlap between them.

We are grateful for your constructive suggestions, which enhanced the overall quality of our research.

3.1. Reasons for selecting gene sets: In this study, we considered the tumor cells and its TME as a whole and tried to completely reconstruct the heterogeneity of the TME. Identifying the components of the TME and their functions, as well as the crosstalk between tumor cells and TME contributes to our understanding of the biological and clinical heterogeneity of CRC. The TME-related signatures included the functional states of the tumor cells, immune and stromal signatures, and metabolic reprogramming features. We selected TME-related signatures only from previous studies, which have demonstrated an association with tumor or a direct link with colorectal cancer. After reviewing previously published studies, the Molecular Signatures Database (MSigDB; http://www.gsea-msigdb.org/gsea/msigdb/index.jsp), and the Reactome pathway portal (https://reactome.org/PathwayBrowser/), we obtained 61 signatures related to tumor, immune, stromal, and metabolic reprogramming features (Supplementary Table2). For instance, signatures such as epithelial-mesenchymal transition (EMT), endothelial cells, and mesenchymal cells were derived from studies related to colorectal cancer (De Sousa et al., 2013; Loboda et al., 2011). (Results section: Establishment of the TME panel)

3.2. Completeness of the gene sets in the context of CRC: To analyze the role of the selected TME-related signatures in colorectal cancer, we performed differential analysis of TME-related signatures abundance between CRC and normal samples, survival analysis, correlation analysis among the TME-related signatures, correlation analysis with TME-related signatures obtained from MCP-counter algorithm, and correlation analysis with the activity of 10 classical oncogenic pathways. (Results section: Establishment of the TME panel) Furthermore, we used the “pamr.predict” function of the R package “pamr” (Tibshirani et al., 2002) to extract centroids of each subtype that best represent each subtype and establish a PAMR classifier. PAM (Prediction Analysis of Microarrays) is a statistical technique to identify subsets of features that best characterize each class using nearest shrunken centroids (Tibshirani *et al.*, 2002). The technique is general and can be used in many other classification problems. As shown in Figure 1—figure supplement 2E, a threshold of 0.566 with minimum 10-fold cross-validation error was selected to identify the 61 TME-related signatures that exhibit at least one non-zero difference between each subtype (seed = 11). (Lines 155-161) These signatures were then used to construct a PAMR classifier with superior predictive capability, exhibiting an overall error rate of 15%. We used the established PAMR classifier to predict the CCCRC subtypes on the CRC-RNAseq cohort and the same four CCCRC subtypes were revealed, with similar patterns of differences in the TME components (Figure 1—figure supplement 2F, G). This indicated that the 61 TME-related signatures best represent each subtype and were indispensable for achieving the identification of the four CCCRC subtypes.

3.3. Any overlap between the gene sets: We have added a list to document the genes that overlap between TME-related signatures (Supplementary file 1b). Supplementary file 1b showed that there was not significant gene overlap between the different signatures.

Collectively, our findings demonstrated that the TME heterogeneity, including unique differences in immune, stromal, and metabolic reprogramming, played a crucial role in tumor development, and that the TME panel could be used to comprehensively characterize CRC.

4) Clarify why clinicopathological covariates were not included.

We are grateful for your astute observations, which enhanced the overall quality of our research. In the initial analysis, we only included the TNM staging variable due to the limited availability of clinical information. We also included the CMS classification system as a covariate in our multivariate Cox proportional hazard regression analysis, given the overlap between our CCCRC classification system and CMS classification system. Our aim was to assess whether the predictive ability of the CCCRC classification system was independent of the CMS classification system. We believe that the addition of the CMS variable has provided important insights into the relationship between these two classification systems.

We have revised our analysis accordingly (Supplementary file 1f, Supplementary file 1g). Specifically, we included additional clinicopathological variables in our multivariate Cox proportional hazard regression analysis, as suggested. These clinicopathological variables included age, gender, tumor site, TNM stage, grade, adjuvant chemotherapy or not, MSI status, BRAF and KRAS mutations, and the CMS classification system. The analysis results showed that the C4 subtype was an independent predictor of the best OS and DFS, whereas the C3 subtype was an independent predictor of the worst OS and DFS after adjustment for the clinicopathological variables in the combined cohort (the CRC-AFFY and CRC-RNAseq cohorts). (Lines 266-271)

5) Clarify the derivation of the single-sample gene classifier, specifying exactly the genes considered and in the final classifier. Please make sure the final classifier is available.

Thank you for your thoughtful review and valuable feedback.

We apologize for any confusion caused in our revised regarding the derivation of the CCCRC classifier. Specifically, we have added more details on the derivation of model genes and the establishment of the model, and ensured the availability of the CCCRC classifier. The method details and results of deriving the model genes and building the model are described next. (Lines 1102-1121) (Lines 562-579)

In order to facilitate the widespread application of CCCRC classification system, we established a simple gene classifier to predict CCCRC subtypes. Firstly, we filtered genes based on their mean expression and variance in the CRC-AFFY cohort, and genes with expression and variance below the bottom 25% were removed. Then, we applied the Random Forest algorithm (RF) in the R package "caret" to perform feature selection on the CCCRC subtype-specific genes of the CRC-AFFY cohort. The top 20 most informative features for each subtype were ranked and selected based on the impurity measure generated by the algorithm. This allowed us to identify critical genes that are strongly associated with each CCCRC subtype and develop the CCCRC classifier. Next, we randomly divided the CRC-AFFY cohort into training and validation sets at a ratio of 7:3 using “createDataPartition” function provided in the R package "caret" (seed=123). The GEP was normalized with Z-scores prior to model training and validation. The CCCRC classifiers were trained with the top 80 subtype-specific genes using the RF, Support Vector Machine (SVM), eXtreme Gradient Boosting (xgboost), and Logistic Regression algorithms implemented in the R package "caret". Finally, we validated the CCCRC classifier on the GSE14333 and GSE17536 datasets, as well as the CRC-AFFY cohort. We evaluated the predictive performance of the CCCRC classifier by evaluating measures such as accuracy value and F1 score, which were generated using the "confusionMatrix" function provided in the R package "caret". (Lines 1102-1121)

We established the CCCRC classifier on the training set by utilizing multiple machine learning algorithms based on the GEP of 80 upregulated subtype-specific genes (Supplementary file 3c). Upon application to the test set, GSE14333, and GSE17536 datasets, the performance of the eXtreme Gradient Boosting (xgboost) algorithm was the best with the highest accuracy values and F1 scores compared to the Random Forest (RF), Support Vector Machine (SVM), and Logistic Regression algorithms (Figure 6—figure supplement 4). Notably, the CCCRC classifier based on the xgboost algorithm displayed robust performance across gene expression platforms, Affymetrix and RNA-sequencing platforms, exhibiting a balanced accuracy of > 80% for all subtypes (Supplementary file 3d). These findings demonstrated the stability and cross-platform applicability of our classifier. The CCCRC classifier based on the xgboost algorithm is publicly available at https://github.com/XiangkunWu/CCCRC_classifier, and the CCCRC subtype information of CRC patients can be obtained by directly inputting the GEP of 80 upregulated subtype-specific mRNA genes. The CCCRC classifier might facilitate the discovery of new biomarkers and the personalized treatment of clinical patients with CRC. (Lines 562-579)

6) To support the main claim that the CCCRC classification can facilitate biomarker discovery, please comment on how best to combine the CCCRC classification together with existing (and future) classifications, to achieve this aim.

Thank you for your thoughtful review and insightful question. We have added a discussion in the article on how to combine existing classification systems (Lines 685-708) and future classification systems (Lines 708-723) to identify new biomarkers and potential therapeutic targets. Overall, extensive research demonstrated a significant correlation between the CCCRC subtype-specific genes and immune evasion mechanisms, underscoring their potential as promising therapeutic targets for developing precision medicine approaches against cancer within the TME. However, realizing the full potential of these classification systems will require a collaborative effort among researchers and clinicians to ensure their effective integration and utilization. Through integration, we can leverage the strengths of different classification schemes to identify new biomarkers and therapeutic targets that can ultimately improve patient outcomes.

Reviewer #1 (Recommendations for the authors):Thank you for your manuscript. Here are my specific suggestions.In Figure S1 A and B, better labels are required. Both seem to be labelled as TCGA. There are open circles that are not labelled.

Thank you very much for your careful review. In response to the reviewer's comment, we have revised the figure labels to provide better clarity and ensure that all elements are accurately identified (Figure 1—figure supplement 1E, F). We have also reviewed the labeling of all other figures in the manuscript to ensure that they are accurate and clear.

Lines 142-144 – Sentence needs expansion. The "prognostic ability" of the signatures was evaluated. The "signatures associated with stromal and tumor compartments significantly correlated".

Thank you very much for your careful review and professional feedback. We have added more content to the revised manuscript based on your suggestions to provide a clearer and more detailed description. The stromal and tumor components significantly were correlated with decreased survival, particularly in the case of mesenchymal cells, endothelial cells, metastasis, differentiation, and EMT signatures. Lymphoid-associated cells generally tended to be associated with a better prognosis, while myeloid-associated cells generally tended to be associated with a poor prognosis. Among the metabolism-related signatures, energy metabolism and carbohydrate metabolism were significantly related to poor prognosis, while nucleotide metabolism, amino acid metabolism, and TCA metabolism were strongly predictive of a favorable prognosis. (Lines 135-142)

Line 152 has the only use of the term "TME panel scores", we presume you mean the signature scores?

Thank you very much for your careful review. We have made the necessary corrections to ensure consistency in the terminology used throughout the manuscript. We have replaced all instances of "TME panel scores" with "TME-related signature scores". (Line 150)

Line 242 contains the first mention of the TCGA-CRC cohort, please reference the supplementary methods.

Thank you very much for your feedback. We have replaced all instances of " TCGA-CRC cohort " with "TCGA-CRC dataset". (Line 286, 291, 890, 896, 910)

Line 642, please use "20X magnification" and "40X magnification", not "20 magnification" or "magnifications".

Thank you very much for your careful review. We have replaced "20 magnification" and "magnifications" with "20X magnification" and "40X magnification" to accurately describe the levels of magnification employed in our study. (Lines 917, 918)

Line 349, "dominate" should be "dominant"?

Thank you very much for your careful review. We have replaced all instances of "dominate" with "dominant". (Lines 411, 419, 437)

Line 1134, I believe this is an UpSet plot, not a Venn plot. (Alexander Lex, Nils Gehlenborg, Hendrik Strobelt, Romain Vuillemot, Hanspeter Pfister. UpSet: Visualization of Intersecting Sets IEEE Transactions on Visualization and Computer Graphics (InfoVis), 20(12): 1983--1992, doi:10.1109/TVCG.2014.2346248, 2014.)

Thank you very much for your professional feedback. We have corrected this issue. (Lines 417)

Figure 6C, D PFS/OS legend is the wrong way around.

Thank you very much for your careful review. We have corrected this issue. (Lines 1569-1570, Figure 6C, D, Legend)

Line 429, Please clarify where the filtered list of 81 genes came from.

Thank you very much for providing such detailed and insightful feedback.

The exact details of the filtering criteria used to obtain the list of pre-clinical model genes have added to the Methods section of the study (Lines 1061-108, Lines 503-511) (Supplementary file 3a). To explore the treatment for each CCCRC subtype using cancer cell line drug-sensitivity experiments, we developed a pre-clinical model based on subtype-specific, cancer cell-intrinsic gene markers according to a previously published study (Eide *et al.*, 2017). Firstly, the “limma” package was used to identify EDGs with FDR < 0.05 between each of the four subtypes and the remaining subtypes in the CRC-AFFY cohort. To identify subtype-specific genes in one of the subtypes, we excluded those that were found to be differentially expressed in comparisons between one of the other subtypes and the remaining subtypes. The upregulated subtype-specific genes (log2 (fold change, FC) > 0 and FDR < 0.05) was ranked based on their log2FC and selected the top 500 genes for further gene screening. Secondly, the GEP of human CRC tissues versus patient-derived xenografts (PDX) in the GSE35144 dataset by the R package “limma” was used to remove those genes associated with stromal and immune components. DEGs with FDR > 0.5 and log2 FC < 1 between human CRC tissues versus PDX were considered as cancer cell-intrinsic genes. Thirdly, we also utilized human CRC cell lines to obtained cancer cell-intrinsic genes. A total of 71 human CRC cell lines with RNAseq data (log2TPM) was obtained from the Genomics of Drug Sensitivity in Cancer (GDSC) database (https://depmap.org/portal/download/all/), 43 of which had dose-response curve (area under the curve, AUC) values. The MSI status, FGA and TMB information of CRC cell lines was obtained from cbioportal website (https://www.cbioportal.org/study/summary?id=ccle_broad_2019). RNAseq data for 71 human CRC cell lines was used to further determine the cancer cell-intrinsic genes and genes among the top 25% within (i) the 10−90 % percentile range of the largest expression values and (ii) the highest expression in at least three samples. The subtype-specific genes and cancer cell-intrinsic genes were intersected to generate the gene list for developing the pre-clinical model. The pre-clinical model was developed using the nearest template prediction (NTP) function of R package “CMScaller”, which can be applied to cross-tissues and cross-platform predictions (Hoshida, 2010). The GEP (log2TPM) of 71 human CRC cell lines normalized by the Z-score were input into the pre-clinical model, and the cell lines were divided into four CCCRC subtypes. (Lines 1061-1088)

Here we want to make a point that we changed from using the xgboost algorithm to using the NTP algorithm to build our pre-clinical model. Based on the genomic features of the cell line, we evaluated the reliability of the final pre-clinical model and found that the pre-clinical model built using the NTP algorithm is more reliable. As expected, the C4 subtype cell lines demonstrated the highest TMB values and MSI frequency while exhibiting the lowest FGA scores when compared to other subtypes (Figure 6—figure supplement 1G-I). In contrast, C1 and C3 subtype cell lines showed significantly higher FGA scores and significantly lower TMB values and MSI frequency. The C2 subtype cell lines had median FGA scores, TMB values, and MSI frequency. The pre-clinical model is publicly available at https://github.com/XiangkunWu/pre_clinical_model. (Lines 503-511)

Lines 452-462, Describe a single-sample gene classifier based on the CRC-RNAseq data with training and a hold-out set. It is also not clear if the 80 gene set was input to the model training or if the final gene list was.

Thank you for your thoughtful review and valuable feedback. We apologize for any confusion caused in our manuscript regarding the derivation of the CCCRC classifier. Specifically, we have added more details on the derivation of model genes and the establishment of the model, and ensured the availability of the CCCRC classifier. We established the CCCRC classifier on the training set by utilizing multiple machine learning algorithms based on the GEP of 80 upregulated subtype-specific mRNA genes (Supplementary file 3c). Upon application to the validation set, the CCCRC classifier displayed robust performance across gene expression platforms, specifically Affymetrix and RNA-sequencing platforms, exhibiting a balanced accuracy of >80% for all subtypes (Supplementary file 3d).

The method details and results of deriving the model genes and building the model are described next. (Lines 1102-1121) (Lines 562-579)

In order to facilitate the widespread application of CCCRC classification system, we established a simple gene classifier to predict CCCRC subtypes. Firstly, we filtered genes based on their mean expression and variance in the CRC-AFFY cohort, and genes with expression and variance below the bottom 25% were removed. Then, we applied the Random Forest algorithm (RF) in the R package "caret" to perform feature selection on the CCCRC subtype-specific mRNA genes of the CRC-AFFY cohort. The top 20 most informative features for each subtype were ranked and selected based on the impurity measure generated by the algorithm. This allowed us to identify critical genes that are strongly associated with each CCCRC subtype and develop the CCCRC classifier. Next, we randomly divided the CRC-AFFY cohort into training and validation sets at a ratio of 7:3 using “createDataPartition” function provided in the R package "caret" (seed=123). The GEP was normalized with Z-scores prior to model training and validation. The CCCRC classifiers were trained with the top 80 subtype-specific genes using the RF, Support Vector Machine (SVM), eXtreme Gradient Boosting (xgboost), and Logistic Regression algorithms implemented in the R package "caret". Finally, we validated the CCCRC classifier on the GSE14333 and GSE17536 datasets, as well as the CRC-AFFY cohort. We evaluated the predictive performance of the CCCRC classifier by evaluating measures such as accuracy value and F1 score, which were generated using the "confusionMatrix" function provided in the R package "caret". (Lines 1102-1121)

We established the CCCRC classifier on the training set by utilizing multiple machine learning algorithms based on the GEP of 80 upregulated subtype-specific mRNA genes (Supplementary file 3c). Upon application to the test set, GSE14333, and GSE17536 datasets, the performance of the eXtreme Gradient Boosting (xgboost) algorithm was the best with the highest accuracy values and F1 scores compared to the Random Forest (RF), Support Vector Machine (SVM), and Logistic Regression algorithms (Figure 6—figure supplement 4). Notably, the CCCRC classifier based on the xgboost algorithm displayed robust performance across gene expression platforms, Affymetrix and RNA-sequencing platforms, exhibiting a balanced accuracy of > 80% for all subtypes (Supplementary file 3d). These findings demonstrated the stability and cross-platform applicability of our classifier. The CCCRC classifier based on the xgboost algorithm is publicly available at https://github.com/XiangkunWu/CCCRC_classifier, and the CCCRC subtype information of CRC patients can be obtained by directly inputting the GEP of 80 upregulated subtype-specific mRNA genes. The CCCRC classifier might facilitate the discovery of new biomarkers and the personalized treatment of clinical patients with CRC. (Lines 562-579)

Line 458: "The gene classifier based on the xgboost algorithm is publicly available at https://github.com/XiangkunWu/CCCRC" but the link does not contain the classifier nor the final list of genes. It has the code used but not the classifier itself. Please upload the classifier in a form that others can use.

Thank you for your thoughtful review and valuable feedback. We apologize for any confusion caused in our manuscript regarding the acquisition of CCCRC classifiers. We have updated the repository to include the CCCRC classifier and final gene list for public use (https://github.com/XiangkunWu/CCCRC_classifier).

Reviewer #2 (Recommendations for the authors):– How robust is the clustering to the choice of clustering method and the choice of gene lists? For example, could the authors comment on what would happen if one of the "tumour cell" signatures is left out, given that "Energy" appears to be a relatively strong component of C2 and C3 for both CRC-AFFY and CRC-RNAseq?This is important to expand on, at least as a Discussion point, because the CCCRC clustering (Figure 1) is a foundational part of this work, and the rest of this paper is structured around these definitions.

Thank you very much for providing such detailed and insightful feedback.

As shown in Figure 1—figure supplement 2E, a threshold of 0.566 with minimum 10-fold cross-validation error was selected to identify the 61 TME-related signatures that exhibit at least one non-zero difference between each subtype (seed = 11). These signatures were then used to construct a PAMR classifier with superior predictive capability, exhibiting an overall error rate of 15%. We used the established PAMR classifier to predict the CCCRC subtypes on the CRC-RNAseq cohort and the same four CCCRC subtypes were revealed, with similar patterns of differences in the TME components (Figure S2F, G). This indicated that the 61 TME-related signatures best represent each subtype and are indispensable for achieving the identification of the four CCCRC subtypes. (Lines 161-168)

1.2. The reviewer has raised a valid concern about the impact of the clustering method selection on the robustness of the clusters.

We performed extensive data analysis attempts during our unsupervised clustering analysis, which primarily involved attempting various clustering methods, including K-means clustering, non-negative matrix factorization (NMF) clustering, and hierarchical clustering, as well as replacing different sources and categories of the TME-related signatures. To determine the optimal clustering method and TME panel, we evaluated whether the TME panel could reproduce the heterogeneity of TME, the stability of the clustering itself, the biological characteristics of the subtypes, the correlation between subtypes and prognosis, and the correlation between subtypes and microsatellite instability (MSI), consensus molecular subtypes (CMS) classification system, and other molecular subtype systems. Due to the abundance of exploratory data analysis results, we ultimately selected the best clustering method and TME panel combination for showcase.

1.2. For example, could the authors comment on what would happen if one of the "tumour cell" signatures is left out, given that "Energy" appears to be a relatively strong component of C2 and C3 for both CRC-AFFY and CRC-RNAseq?

We added a sensitivity analysis of the effect of TME-related signature on the clustering results in revised manuscript, as detailed in the first question of Public Review. Limiting the consensus clustering analysis to only immune-related or immune- and stroma-related signatures did not allow reliable identification of all four CCCRC subtypes, highlighting the necessity of our carefully designed TME panel.

– Given that the 61 gene signatures appear to clearly separate CCR-AFFY and CRC-RNAseq into 4 clusters using the presented method (Figure 1A), could the authors comment on why the CCCRC and CMS subtypes were not more similar to each other (Figure 1G)? Is there a component captured by (or driving) CCCRC which is not captured by CMS, and vice versa?

Thank you for this insightful comment. The consensus molecular subtype (CMS) integrated six independent classification systems based on gene expression profiles (CMS1: MSI immune subtype; canonical; CMS3: metabolic subtype; CMS4: mesenchymal subtype). Each of these classification systems was derived from gene expression profiles of different datasets and employs different unsupervised clustering methods. The molecular subtypes defined by these six classification systems generally include distinct immune subtypes, such as the inflammatory subtype in the Sadanandam classification system, the mesenchymal subtype and CIMP-H-like subtype in the Budinska classification system. Therefore, we considered that the subtype features contained in the six independent classification systems relied upon by the CMS classification system fundamentally cover the information contained in the TME-related signatures we have collected.

Nevertheless, we found that the CCCRC classification system, which was developed by gathering TME-related signatures based on prior knowledge and applying consensus clustering methods, could better elucidate the heterogeneity of the tumor microenvironment (C1: proliferative subtype; C2: immunosuppressed subtype; C3: immune-excluded subtype; C4: immunomodulatory subtype). We believe that this is the main reason for the classification differences between the CCCRC classification system and the CMS classification system. Meanwhile, we found that the C1, C2/C3, and C4 subtypes of the CCCRC system exhibited significant overlap with the CMS2, CMS4, and CMS1 subtypes, respectively. Notably, CMS3 is a metabolic reprogramming-related molecular subtype lacking an immune subtype counterpart, predominantly composed of C1, C2, and C4 subtypes. Furthermore, the C1 subtype is less immunogenic than CMS2 and more closely resembles cold tumor characteristics. Compared with the CMS1 subtype, the C4 subtype showed upregulated anti-tumor-immune components and lacked immunosuppressive components. Specifically, we subdivided the CMS4 subtype into the C2 and C3 subtypes with distinct molecular and clinical features. In terms of the TME components, the C2 subtypes within CMS4 exhibit a higher abundance of immune components, such as T cells and NK cells, compared to the C3 subtype within CMS4. However, the differences in stromal components between these subtypes were not statistically significant. In terms of the clinical features, the C2 subtype within the CMS4 subtype also had a better prognosis than the C3 subtype within the CMS4 subtype. Our findings indicate that the CCCRC classification system overlaps with the CMS classification system in part, and our classification system can further refine the CMS classification system, which helps to elucidate the correlation between CMS subtypes and tumor microenvironment heterogeneity. (Lines 222-255, 656-685)

– In addition to the summary in Fig1G, it would be helpful if Figure 1A heatmap was also annotated with CMS and MSI annotations, reordering the samples within each CCCRC cluster if necessary.

Thank you very much for providing such detailed feedback. We have added CMS and MSI annotations to Figure 1A heatmap and reordered the samples within each CCCRC cluster to better illustrate the relationships between them.

– The completeness (or otherwise) of the gene lists chosen for the gene panel should be made more explicit. To what extent is the chosen gene panel likely to represent the components of CRC tumour-associated immune cells, tumour cells, etc? Does the panel of gene signatures capture all the biological differences previously described between CMS subtypes, e.g. TGFb activation; WNT and MYC activation; others?

The reviewer's feedback was invaluable and greatly improved the quality of our research.

4.1. The completeness (or otherwise) of the gene lists chosen for the gene panel should be made more explicit.

In this study, we considered the tumor cells and its TME as a whole and tried to completely reconstruct the heterogeneity of the TME. Identifying the components of the TME and their functions, as well as the crosstalk between tumor cells and TME contributes to our understanding of the biological and clinical heterogeneity of CRC. The TME-related signatures included the functional states of the tumor cells, immune and stromal signatures, and metabolic reprogramming features. Regarding the source of TME-related signature, we carefully selected only those signatures that have been demonstrated in previous studies to be associated with tumor or directly related to colorectal cancer. After reviewing previously published studies, the Molecular Signatures Database (MSigDB; http://www.gsea-msigdb.org/gsea/msigdb/index.jsp), and the Reactome pathway portal (https://reactome.org/PathwayBrowser/), we obtained 61 signatures related to tumor, immune, stromal, and metabolic reprogramming features (Supplementary Table2). For example, signatures such as epithelial-mesenchymal transition (EMT), endothelial cells, and mesenchymal cells were derived from studies related to colorectal cancer (De Sousa *et al.*, 2013; Loboda *et al.*, 2011). (Results section: Establishment of the TME panel)

4.2. To what extent is the chosen gene panel likely to represent the components of CRC tumour-associated immune cells, tumour cells, etc?

To analyze the role of the selected TME-related signatures in colorectal cancer, we performed differential analysis of TME-related signatures abundance between CRC and normal samples, survival analysis, correlation analysis among the TME-related signatures, and correlation analysis with TME-related signatures obtained from MCP-counter algorithm. (Results section: Establishment of the TME panel) The final clustering results also demonstrated the reliability of the selected TME-related signatures: analysis of the differences in the abundance of TME-related signatures obtained by other algorithms (CIBERSOFT, MCP-counter and ESTIMATE algorithms) among the four subtypes, differences in 10 classical oncogenic pathway scores among the four subtypes, as well as differences in the distribution and function of T cell subpopulations among the four subtypes. We have added a list to document the genes that overlap between TME-related signatures (Supplementary file 1b).

4.3. Does the panel of gene signatures capture all the biological differences previously described between CMS subtypes, e.g. TGFb activation; WNT and MYC activation; others?

When collecting TME-related signatures, we focused on TME-related signatures associated with cancer or colorectal cancer, including the functional states of tumor cells, immune and stromal components, and metabolic reprogramming signatures. We have added analyses of the correlation between 15 signatures associated with the functional states of tumor cells and the activity of 10 classical oncogenic pathways, such as TGFb activation, WNT activation, and MYC activation (Figure 1—figure supplement 1K). Our findings show a significant positive correlation between them. In addition, our findings demonstrated a strong positive correlation between lymphocytic and stromal signatures and MCP-counter algorithm-derived signatures (Figure 1—figure supplement 1I, J).

The biological characteristics of CMS subtypes are also mainly reflected in tumor cell-associated features, as well as immune, stromal, and metabolic characteristics. Our TME panel comprehensively captures features of both tumor cells and TME. Therefore, we believe that our TME panel captures the biological characteristics relevant to CMS subtypes.

– What is the overlap in samples between CRC-AFFY and those used for the development of the CMS classification, and/or the original papers which fed into the CMS? Are all of these cohorts similar to each other, or overlapping? Was there a similar balance of subtypes and clinicopathological variables? This would help the reader to understand all possible sources of the difference between CCCRC and the current consensus CMS.

We greatly appreciate the reviewer's suggestions and believe that they are crucial in improving the quality of our paper. The transcriptomic data used to identify CCCRC subtypes in this study was essentially the same as the cohort used to develop CMS subtypes (Guinney et al., 2015). The transcriptomic datasets used to establish the CCCRC classification system in this study included GSE13067, GSE13294, GSE14333, GSE17536, GSE33113, GSE37892, GSE38832, GSE39582, TCGA-COAD/READ, CPTAC-colon/rectum. The transcriptomic datasets used to establish the CMS classification system in this study included GSE2109, GSE13067, GSE13294, GSE14333, GSE17536, GSE20916, GSE23878, GSE37892, GSE39582, GSE33113, TCGA-COAD/READ. Therefore, the heterogeneity of the two classification systems caused by the source heterogeneity of the transcriptomic datasets can be excluded.

We further analyzed the association of CCCRC subtypes with clinicopathological characteristics (Supplementary file 1g). We found that the C4 subtype was mostly diagnosed in right-sided CRC lesions and in females, which was consistent with the CMS1 subtype. The C1 and C3 subtypes were mainly observed in left-sided CRC lesions and in males, consistent with the CMS2 and CMS4 subtypes. The C3 subtype was strongly associated with more advanced tumor stages, which was the similarity to the CMS4 subtype, while the C4 subtype was associated with higher histopathological grade, which was the similarity to the CMS1 subtype.

– Please clarify why clinicopathological covariates were not included in the OS and PFS models (Figures1, 4-6). For example, left-sidedness, KRAS or BRAF mutations, MSI; e.g. see Guinney (2015) Table S13.

We are grateful for the reviewer's astute observations, which enhanced the overall quality of our research. In the initial analysis, we only included the TNM staging variable due to the limited availability of clinical information. We also included the CMS classification system as a covariate in our multivariate Cox proportional hazard regression analysis, given the overlap between our CCCRC classification system and CMS classification system. Our aim was to assess whether the predictive ability of the CCCRC classification system was independent of the CMS classification system. We believe that the addition of the CMS variable has provided important insights into the relationship between these two classification systems.

We appreciate your valuable suggestions, and we have revised our analysis accordingly (Supplementary file 1f). Specifically, we have included additional clinicopathological variables in our multivariate Cox proportional hazard regression analysis, as suggested. These clinicopathological variables include age, gender, tumor site, TNM stage, grade, adjuvant chemotherapy or not, MSI status, BRAF and KRAS mutations, and the CMS classification system. We believe that the addition of these variables has strengthened our analysis and improved the overall quality of our manuscript.

Multivariate Cox proportional hazard regression analysis showed that the C4 subtype was an independent predictor of the best OS and DFS, whereas the C3 subtype was an independent predictor of the worst OS and DFS after adjustment for the clinicopathological variables in the combined cohort (the CRC-AFFY and CRC-RNAseq cohorts). (Lines266-271)

– Fig1I: "Forest plot of multivariate Cox proportional hazard regression analysis of PFS after adjusting for TNM stage and CMS subtype in the CRC-AFFY cohort." – is this intended to mean that CMS was a covariate? If yes, why?

Yes, it indicates that CMS was included as a covariate in the analysis. In the initial analysis, we only included the TNM staging variable due to the limited availability of clinical information. We also included the CMS classification system as a covariate in our multivariate Cox proportional hazard regression analysis, given the overlap between our CCCRC classification system and CMS classification system. Our aim was to assess whether the predictive ability of the CCCRC classification system was independent of the CMS classification system. We believe that the addition of the CMS variable has provided important insights into the relationship between these two classification systems.

– The discussion comparing the reported predictive or prognostic value of CCCRC subtypes (this work) and CMS subtypes (published) could be expanded. For which endpoints did the CMS subtypes (published) have superior or similar separation than the CCCRC classification (this work), and does this affect the biological interpretation of each classification scheme?

The reviewer's feedback was invaluable and greatly improved the quality of our research. The comparison between the predictive or prognostic value of CCCRC subtypes and CMS subtypes could indeed be expanded upon. Considering that the C1, C2/C3, and C4 subtypes partially overlap with the CMS2, CMS4, and CMS1 subtypes, respectively, we further analyzed the prognostic differences between them in the combined cohort. We observed that patients with the CMS1 subtype, characterized by high immune infiltration and activation, did not have the best prognosis compared with the other CMS subtypes, while patients with the CMS2 subtype, characterized by low immune infiltration, had the best prognosis (Becht et al., 2016; Guinney *et al.*, 2015). This may be due to the fact that CMS1 subtype has a higher accumulation of fibroblasts compared to CMS2 subtype (Guinney *et al.*, 2015). Moreover, this study has revealed that, in comparison with the C4 subtype, the CMS1 subtype has fewer anti-tumor immune components, more stroma components, and other immunosuppressive components, resulting in a poorer prognosis. The prognosis of patients with the C2 subtype was also better than that of patients with the CMS4 and C3 subtypes, and the C2 subtype within the CMS4 subtype also had a better prognosis than the C3 subtype within the CMS4 subtype, which might be related to the presence of more immune infiltration in the C2 subtype than in the C3 and CMS4 subtypes. Our CCCRC classification system further subdivided CMS4 into two subtypes, the C2 and C3 subtypes, each with distinct molecular and prognostic characteristics. Meanwhile, the C4 subtype with MSI exhibited higher immune infiltration and less stromal component compared with the CMS4 subtype with MSI. Overall, our CCCRC classification system can be combined with the CMS classification system and other molecular subtypes to enhance the understanding of the association between TME heterogeneity and different clinical phenotypes. (Lines 271-284,656-685)

– The claims "the C4 subtype was suitable for ICB treatment" and "significance of CCCRC in guiding the clinical treatment of colorectal cancer", and any similar phrases about guiding treatment, need to be rephrased; this is a pre-clinical study.

Thank you for your insightful feedback. We agree that our retrospective analysis of the CCCRC classification in relation to disease progression under immune checkpoint blockade treatment does not directly support clinical treatment decisions. We acknowledge that additional experimental evidence would be required to fully support the use of the CCCRC classification as a clinical tool for guiding treatment decisions. We have highlighted in the corresponding section of the article that this research is pre-clinical and still requires substantial basic experiments and clinical trials to validate. (Lines 536, 751)

– To support the main claim that the CCCRC classification can facilitate biomarker discovery, can the authors comment on how best to combine the CCCRC classification together with existing (and future) classifications, to achieve this aim?

We greatly appreciate the reviewer's invaluable comments, which have significantly improved the quality of our research. We have added a discussion in the article on how to combine existing classification systems (Lines 686-709) and future classification systems (Lines 709-724) to identify new biomarkers and potential therapeutic targets. Overall, extensive research has demonstrated a significant correlation between the CCCRC subtype-specific genes and immune evasion mechanisms, underscoring their potential as promising therapeutic targets for developing precision medicine approaches against cancer within the TME. However, realizing the full potential of these classification systems will require a collaborative effort among researchers and clinicians to ensure their effective integration and utilization. Through integration, we can leverage the strengths of different classification schemes to identify new biomarkers and therapeutic targets that can ultimately improve patient outcomes.

– Could expand the Discussion on whether/how this "holistic" CCCRC classification might contribute towards understanding the heterogeneity in treatment outcomes seen in different CRC trials (e.g. FIRE-3, COIN, CALGB/SWOG 80405).

The reviewer's comments greatly contributed to the quality of our study. The CCCRC classification system also facilitates the understanding of the differences in the results of the CALGB/SWOG 80405 and FIRE-3 clinical trials(Heinemann *et al.*, 2014; Lenz *et al.*, 2019). Both trials evaluated the combination of cetuximab or bevacizumab with a different chemotherapy backbone: in the CALGB/SWOG 80405 trial, 75% of patients received oxaliplatin, while in the FIRE-3 trial, all patients received irinotecan. Aderka et al. discussed the reasons for the differences in the results of these two clinical trials, which may be related to differences in the chemotherapy backbone used and TME heterogeneity (Aderka et al., 2019). Based on our examination of the results summarized in Figure 4 of the work by Aderka et al. (Aderka et al., 2019), we found that differences in the treatment outcomes of the CMS1 and CMS4 subtypes were the crucial factor behind the divergent results observed in the two clinical trials. The CMS1 and CMS4 subtypes have a microenvironment rich in CAFs. Our CCCRC classification results also showed that CMS1, in addition to mainly consisting of the C4 subtype, also contains a considerable number of the C2 subtype, while the CMS4 subtype mainly consists of the C2 and C3 subtypes. Furthermore, our study results indicated that the C2 subtype is suitable for chemotherapy in combination with bevacizumab, possibly because the combination can inhibit the CAFs and abnormal blood vessel formation in the TME, thus alleviating the immune suppression of the immune cells. However, the C3 subtype is not suitable for chemotherapy in combination with bevacizumab because it only accumulates CAFs and abnormal blood vessel formation but lacks T cell infiltration. Therefore, we boldly speculate that the CMS1 and CMS4 subtypes in the CALGB/SWOG 80405 clinical trial may contain more C2 subtypes than those in the FIRE-3 clinical trial, leading to the CMS1 and CMS4 subtypes in the CALGB/SWOG 80405 clinical trial being more suitable for chemotherapy in combination with bevacizumab than cetuximab compared to the FIRE-3 clinical trial. Overall, the integration of CCCRC and CMS classification systems provides valuable insights for understanding the divergent outcomes of the two clinical trials. (Lines 725-752)

Reviewer #3 (Recommendations for the authors):The authors' title "histopathology-molecular analysis" suggests the manuscript includes an emphasis on the investigation of histopathological parameters. Whilst the authors do examine the relationship between histopathological analysis of specimens with CCCRC subtypes this features as a minor aspect of the study which largely focuses on large-scale investigation of transcriptional signatures.Figures1C, 2C, 3B and 4E – pvalues on plots illegible

Thank you for bringing to our attention the issue regarding the illegibility of p-values on Figures. We have revised the figures and increased the font size of the p-values to ensure their legibility. (Figure 1C, 2C, 3B and 4DF)

Methodology for tumour evolution patterns needs comment/expansion – I could not identify this in the current methods.

Thank you for your insightful comments on the method section of our manuscript. We appreciate your feedback and have revised the section to provide more information on the analysis of tumor evolution patterns.

We performed differential analysis between each of the four subtypes and the remaining subtypes using Wilcoxon rank-sum test to identify DMGs with FDR < 0.001 and differentially expressed proteins (DEPs) with *P*-value < 0.05. The difference of DMGs and DEPs was defined as the difference between each of the four subtypes and the remaining subtypes. The “limma” package was used to identify differentially expressed genes (DEGs) with FDR < 0.001 between each of the four subtypes and the remaining subtypes. To identify subtype-specific DMGs, DEPs and DEGs in one of the subtypes, we excluded those that were found to be differentially expressed in comparisons between one of the other subtypes and the remaining subtypes. Additionally, we performed differential analysis of methylation gene, gene expression, and protein levels between each subtype based on the above method (Lines 983-994).

According to the theory of linear tumor evolution, a dominant evolutionary pattern would involve gradual changes in certain molecular features across subtypes, either increasing or decreasing (Davis et al., 2017; Tavernari et al., 2021). To begin with, we obtained the common set of all DMGs (FDR < 0.05), DEGs (FDR < 0.05) and DEPs (P-value < 0.5) between each subtype, including both upregulated and downregulated ones, by taking their intersection. Next, we separately intersected the sets of upregulated and downregulated DMGs, DEGs and DEPs among each subtype in the common set to explore potential evolutionary patterns. From the UpSet plot (Lex et al., 2014), it is evident that the evolutionary pattern from C1 to C4, followed by C2, and C3 subtypes had the same sign in difference or log2FC and were dominant: either all positive for increasing DNA methylation/gene expression/protein levels (red), or all negative for decreasing DNA methylation/gene expression/protein levels (blue) (Figure 4A-C). For instance, the top DEGs SERPINH1 and NDUFA10 showed a gradual increase and decrease in expression from C1, C4, C2 to C3 subtypes, respectively (Figure 4D). (Lines 411-423).

Comment on the degree of intra-patient heterogeneity of CCCRC subtypes would be nice. A significant degree of overlap between immunosuppressive and immune excluded, and proliferative and immuno-modulatory signatures in Figure 1A is apparent and should be commented upon.

Thank you very much for providing such detailed and insightful feedback.

3.1. Comment on the degree of intra-patient heterogeneity of CCCRC subtypes would be nice.

We have added intra-tumor heterogeneity analysis for each subtype (Lines 196-198). The level of intratumor heterogeneity (ITH) was significantly linked to poor prognosis and drug resistance (Caswell and Swanton, 2017). The ITH data used in our study for the CRC-RNAseq cohort was obtained from a previous study conducted by Thorsson et al. (Thorsson *et al.*, 2018). As expected, the ITH of the C2 and C3 subtypes was higher than that of the other subtypes, while the ITH of the C4 subtype was the lowest (Figure 1F). Our analysis using the Kaplan-Meier method demonstrated that patients with the C4 subtype had significantly higher overall survival (OS) and disease-free survival (DFS) compared to those with the C2 and C3 subtypes. Furthermore, the C3 subtype was resistant to chemotherapy, cetuximab, bevacizumab, and ICB therapy. Our investigation of drug sensitivity data of cell lines also indicated that the C2 and C3 subtypes were generally not responsive to most drugs.

3.2. A significant degree of overlap between immunosuppressive and immune excluded, and proliferative and immuno-modulatory signatures in Figure 1A is apparent and should be commented upon.

Our research revealed that both C2 and C3 subtypes exhibited a high level of tumor stroma, while C1 and C4 subtypes were characterized by active DNA damage and repair and high tumor proliferation. Additionally, C2 and C4 subtypes had an abundance of immune components. This was consistent with our finding that there may be interconversion between the C1 and C4 subtypes, between the C4 and C2 subtypes, and between the C2 and C3 subtypes in this evolutionary pattern. The interconversion between C2 and C4 subtypes in this evolutionary pattern was the rarest situation, indicating that once the tumor enters the C2 subtype, it is difficult to reverse and will progress to the C3 subtype. (Lines 637-644)

The addition of a spatially resolved RNA seq dataset (likely available from Nanostring of 10X genomics Visium platforms) would be nice to see the distribution of CCCRC subtypes within each patient.

We are grateful for the reviewer's astute observations, which enhanced the overall quality of our research.

To investigate the spatial distribution relationship between four CCCRC subtypes of tumor cells, T cells, and stromal cells, we conducted a re-analysis of publicly available CRC spatial transcriptomics data (ST) obtained from the 10X website (https://www.10xgenomics.com/resources/datasets). The Space Ranger output files were then processed with Seurat (V4.1.1) (Hao *et al.*, 2021) using SCTransform for normalization (Hafemeister and Satija, 2019). RunPCA were used to dimension reduction and RunUMAP to visualize the data. We used “ssGSEA” method implemented in the R package “GSVA” to score the six cell types (C1-C4 subtype cancer cells, mesenchymal cells, and T cells) (Hänzelmann *et al.*, 2013). The “ssGSEA” method has been previously demonstrated to be highly reliable and suitable for ST data analysis (Wu *et al.*, 2022). The cell-type-rich region was defined as the ssGSEA score of each cell type from one spot larger than the 75% quantile of this cell type. The markers for the six cell types are listed in the Supplementary file 1a and Supplementary file 3a. (Lines 1090-1102)

The Cytassist and Visium samples had a total of 9080 and 2660 spots, respectively. We used “ssGSEA” method to quantify the six cell subpopulations of each spot and also visualized only the spots corresponding to the top 25% of the score ranking for each cell type (Figure 6—figure supplement 2AB, Figure 6—figure supplement 3AB). In Cytassist samples, we observed different spatial distribution patterns of the four subtypes of tumor cells (Figure 6—figure supplement 2B). Specifically, the C3 subtype of tumor cells was predominantly located in the tumor periphery with an enrichment of mesenchymal cells and T cells (areas selected by black dashed circles). In contrast, the C4 subtype of tumor cells was mainly present in the center of the tumor, accompanied by the presence of T cells. The C1 and C2 subtypes of tumor cells were distributed in relatively uniform areas, mainly in the tumor periphery, with fewer mesenchymal cells and T cells. However, the distribution areas of C2 subtype and C3 subtype of tumor cells also partially were in overlap (the area selected by red dashed circles). The same distribution patterns can also be observed in the Visium sample (Figure 6—figure supplement 3B). Further analysis of the correlation between the ssGSEA scores of each cell type in the cell-type-rich regions and those of other cell types was conducted (Figure 6—figure supplement 2D, E, Figure 6—figure supplement 3D, E). We found that in the C3 subtype-rich region of tumor cells, the C3 subtype score of tumor cells was significantly positively correlated with the mesenchymal cell score, while in the T cell-rich region, the C3 subtype score of tumor cells was significantly negatively correlated with the T cell score. The C4 subtype score of tumor cells was significantly positively correlated with the T cell score and negatively correlated with the mesenchymal cell score in the C4 subtype-rich, T cell-rich, and mesenchymal cell-rich regions. The C1 subtype and C2 subtype scores of tumor cells were negatively correlated with mesenchymal cell and T cell scores. Overall, these results were generally consistent with previous histopathologic analysis findings. (Lines 538-562)

[Editors' note: further revisions were suggested prior to acceptance, as described below.]

The manuscript has been improved but there are some remaining issues that need to be addressed, as outlined below:1) Please remove the word "histopathology" from the title since that analysis was not a major driver of the results.

We are grateful for your constructive suggestions. We have removed the word "histopathology" from the title, as it was not a major driver of the results.

2) The limitations of the study have been added as requested. Could you please add a Limitations paragraph in the Discussion, to summarise all the study's limitations in one place.

We have added a Limitations paragraph in the Discussion section to summarize all the study's limitations in one place. (Lines 807-813)

3) Please add to the limitations paragraph that the removal of single signatures at the clustering stage was not evaluated. From the Response (page 13): "Since the effect of removing one of the TME-related signatures on the clustering results was not well evaluated, we attempted to remove the entire category."

We have included a statement in the Limitations paragraph stating that the effect of removing single TME-related signatures on the clustering results was not well evaluated, and that we attempted to remove the entire category instead. (Lines 809-811)

4) Statements about clinical decisions still remain, for example:Line 753: "The CCCRC classification system may facilitate clinical treatment decisions."Line 757: "We demonstrated the suitability of C4 for immune checkpoint therapy".and in other places.Given the study limitations, all such statements should be toned down.e.g. "in future … may contribute to.. subject to further work.. in combination with other classification systems.."

We have toned down statements regarding clinical decisions, as suggested, and instead focused on the potential of our CCCRC classification system to contribute to the design of future clinical trials. (Lines 752-759)

References

Aderka, D., Stintzing, S., and Heinemann, V. (2019). Explaining the unexplainable: discrepancies in results from the CALGB/SWOG 80405 and FIRE-3 studies. The Lancet. Oncology *20*, e274-e283. 10.1016/s1470-2045(19)30172-x.

Bagaev, A., Kotlov, N., Nomie, K., Svekolkin, V., Gafurov, A., Isaeva, O., Osokin, N., Kozlov, I., Frenkel, F., Gancharova, O., et al. (2021). Conserved pan-cancer microenvironment subtypes predict response to immunotherapy. Cancer cell *39*, 845-865.e847. 10.1016/j.ccell.2021.04.014.

Becht, E., de Reyniès, A., Giraldo, N.A., Pilati, C., Buttard, B., Lacroix, L., Selves, J., Sautès-Fridman, C., Laurent-Puig, P., and Fridman, W.H. (2016). Immune and Stromal Classification of Colorectal Cancer Is Associated with Molecular Subtypes and Relevant for Precision Immunotherapy. Clinical cancer research : an official journal of the American Association for Cancer Research *22*, 4057-4066. 10.1158/1078-0432.Ccr-15-2879.

Caswell, D.R., and Swanton, C. (2017). The role of tumour heterogeneity and clonal cooperativity in metastasis, immune evasion and clinical outcome. BMC medicine *15*, 133. 10.1186/s12916-017-0900-y.

Davis, A., Gao, R., and Navin, N. (2017). Tumor evolution: Linear, branching, neutral or punctuated? Biochimica et biophysica acta. Reviews on cancer *1867*, 151-161. 10.1016/j.bbcan.2017.01.003.

De Sousa, E.M.F., Wang, X., Jansen, M., Fessler, E., Trinh, A., de Rooij, L.P., de Jong, J.H., de Boer, O.J., van Leersum, R., Bijlsma, M.F., et al. (2013). Poor-prognosis colon cancer is defined by a molecularly distinct subtype and develops from serrated precursor lesions. Nature medicine *19*, 614-618. 10.1038/nm.3174.

Eide, P.W., Bruun, J., Lothe, R.A., and Sveen, A. (2017). CMScaller: an R package for consensus molecular subtyping of colorectal cancer pre-clinical models. Scientific reports *7*, 16618. 10.1038/s41598-017-16747-x.

Guinney, J., Dienstmann, R., Wang, X., de Reyniès, A., Schlicker, A., Soneson, C., Marisa, L., Roepman, P., Nyamundanda, G., Angelino, P., et al. (2015). The consensus molecular subtypes of colorectal cancer. Nature medicine *21*, 1350-1356. 10.1038/nm.3967.

Hafemeister, C., and Satija, R. (2019). Normalization and variance stabilization of single-cell RNA-seq data using regularized negative binomial regression. Genome biology *20*, 296. 10.1186/s13059-019-1874-1.

Hänzelmann, S., Castelo, R., and Guinney, J. (2013). GSVA: gene set variation analysis for microarray and RNA-seq data. BMC bioinformatics *14*, 7. 10.1186/1471-2105-14-7.

Hao, Y., Hao, S., Andersen-Nissen, E., Mauck, W.M., 3rd, Zheng, S., Butler, A., Lee, M.J., Wilk, A.J., Darby, C., Zager, M., et al. (2021). Integrated analysis of multimodal single-cell data. Cell *184*, 3573-3587.e3529. 10.1016/j.cell.2021.04.048.

He, Y., Jiang, Z., Chen, C., and Wang, X. (2018). Classification of triple-negative breast cancers based on Immunogenomic profiling. Journal of experimental & clinical cancer research : CR *37*, 327. 10.1186/s13046-018-1002-1.

Heinemann, V., von Weikersthal, L.F., Decker, T., Kiani, A., Vehling-Kaiser, U., Al-Batran, S.E., Heintges, T., Lerchenmüller, C., Kahl, C., Seipelt, G., et al. (2014). FOLFIRI plus cetuximab versus FOLFIRI plus bevacizumab as first-line treatment for patients with metastatic colorectal cancer (FIRE-3): a randomised, open-label, phase 3 trial. The Lancet. Oncology *15*, 1065-1075. 10.1016/s1470-2045(14)70330-4.

Hoshida, Y. (2010). Nearest template prediction: a single-sample-based flexible class prediction with confidence assessment. PloS one *5*, e15543. 10.1371/journal.pone.0015543.

Lenz, H.J., Ou, F.S., Venook, A.P., Hochster, H.S., Niedzwiecki, D., Goldberg, R.M., Mayer, R.J., Bertagnolli, M.M., Blanke, C.D., Zemla, T., et al. (2019). Impact of Consensus Molecular Subtype on Survival in Patients With Metastatic Colorectal Cancer: Results From CALGB/SWOG 80405 (Alliance). Journal of clinical oncology : official journal of the American Society of Clinical Oncology *37*, 1876-1885. 10.1200/jco.18.02258.

Lex, A., Gehlenborg, N., Strobelt, H., Vuillemot, R., and Pfister, H. (2014). UpSet: Visualization of Intersecting Sets. IEEE transactions on visualization and computer graphics *20*, 1983-1992. 10.1109/tvcg.2014.2346248.

Loboda, A., Nebozhyn, M.V., Watters, J.W., Buser, C.A., Shaw, P.M., Huang, P.S., Van't Veer, L., Tollenaar, R.A., Jackson, D.B., Agrawal, D., et al. (2011). EMT is the dominant program in human colon cancer. BMC medical genomics *4*, 9. 10.1186/1755-8794-4-9.

Tavernari, D., Battistello, E., Dheilly, E., Petruzzella, A.S., Mina, M., Sordet-Dessimoz, J., Peters, S., Krueger, T., Gfeller, D., Riggi, N., et al. (2021). Nongenetic Evolution Drives Lung Adenocarcinoma Spatial Heterogeneity and Progression. Cancer discovery *11*, 1490-1507. 10.1158/2159-8290.Cd-20-1274.

Thorsson, V., Gibbs, D.L., Brown, S.D., Wolf, D., Bortone, D.S., Ou Yang, T.H., Porta-Pardo, E., Gao, G.F., Plaisier, C.L., Eddy, J.A., et al. (2018). The Immune Landscape of Cancer. Immunity *48*, 812-830.e814. 10.1016/j.immuni.2018.03.023.

Tibshirani, R., Hastie, T., Narasimhan, B., and Chu, G. (2002). Diagnosis of multiple cancer types by shrunken centroids of gene expression. Proceedings of the National Academy of Sciences of the United States of America *99*, 6567-6572. 10.1073/pnas.082099299.

Wu, Y., Yang, S., Ma, J., Chen, Z., Song, G., Rao, D., Cheng, Y., Huang, S., Liu, Y., Jiang, S., et al. (2022). Spatiotemporal Immune Landscape of Colorectal Cancer Liver Metastasis at Single-Cell Level. Cancer discovery *12*, 134-153. 10.1158/2159-8290.Cd-21-0316.